# FeatSharp: Your Vision Model Features, Sharper

Mike Ranzinger [1] [*]    Greg Heinrich [1] [*]    Pavlo Molchanov [1]    Bryan Catanzaro [1]    Andrew Tao [1]

## Abstract

The feature maps of vision encoders are fundamental to myriad modern AI tasks, ranging from core perception algorithms (e.g. semantic segmentation, object detection, depth perception, etc.) to modern multimodal understanding in vision-language models (VLMs). Currently, in computer vision, the frontier of general purpose vision backbones is Vision Transformers (ViT), typically trained using contrastive loss (e.g. CLIP). A key problem with most off-the-shelf ViTs, particularly CLIP, is that these models are inflexibly low resolution. Most run at $224 \times 224$px, while the "high-resolution" versions are around $378 - 448$px, but still inflexible. We introduce a novel method to coherently and cheaply upsample the feature maps of low-resolution vision encoders while picking up on fine-grained details that would otherwise be lost due to resolution. We demonstrate the effectiveness of this approach on core perception tasks as well as within agglomerative model training using RADIO as a way of providing richer targets for distillation. Code available at https://github.com/NVlabs/FeatSharp.

## 1. Introduction

Vision foundation models (VFM) (Awais et al., 2023) have seen widespread use since the beginning of the modern era of computer vision using deep learning (Krizhevsky et al., 2012), primarily used to perform transfer learning (Plested & Gedeon, 2022) (e.g. finetuning a VFM on a downstream task), information retrieval (Babenko et al., 2014; Zhang & Liu, 2024), and most recently, to power visual capabilities for vision-language models (VLM) (Alayrac et al., 2022; et al., 2024; Liu et al., 2023; Lin et al., 2023). The recent shift toward using Transformers (Vaswani et al., 2017) for computer vision (ViT (Dosovitskiy et al., 2021)) has tremendously moved the field forward, but has generally

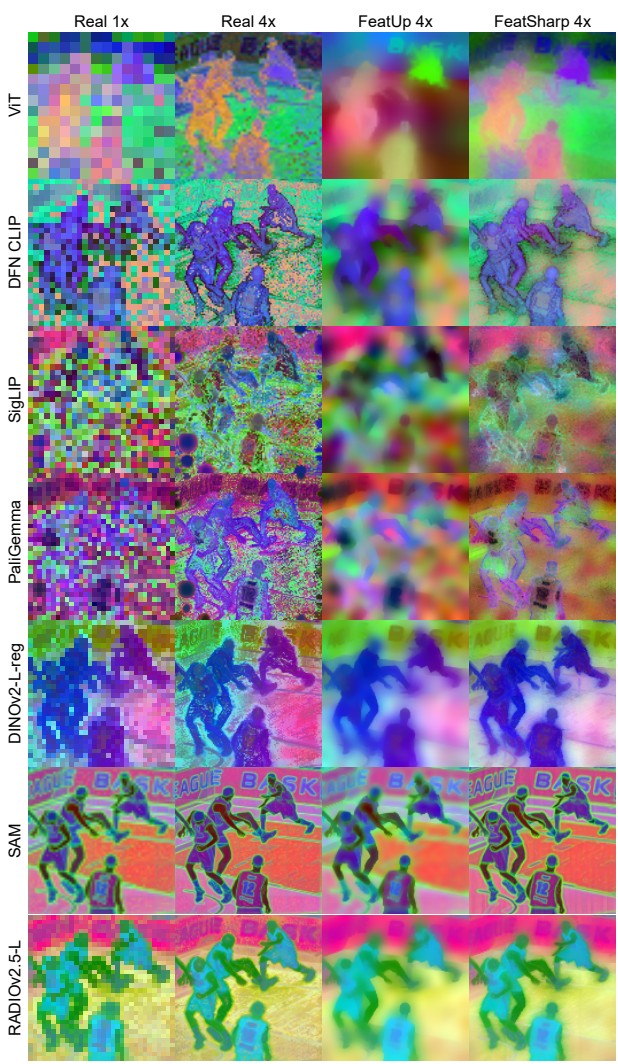

Figure 1. PCA visualizations of features from a basketball scene. **Column 1**: Raw features produced by the model at normal resolution (e.g. 14x downsample for DFN CLIP, SigLIP, PaliGemma, and DINOv2, 16x downsample for SAM and RADIOv2.5-L. **Column 2**: Raw features at the 4x upsample resolution (we interpolate the position embeddings for those models that don't natively support resolution changes). **Column 3**: FeatUp-JBU 4x upsampling (*prior work*). **Column 4**: FeatSharp 4x upsampling.
*NOTE: "Real 4x" technically only makes sense for models with strong scale equivariance, such as DINOv2, RADIO, and SAM.*

---

[*]Equal contribution  [1]NVIDIA. Correspondence to: Mike Ranzinger <mranzinger@nvidia.com>.

*Proceedings of the 42nd International Conference on Machine Learning*, Vancouver, Canada. PMLR 267, 2025. Copyright 2025 by the author(s).

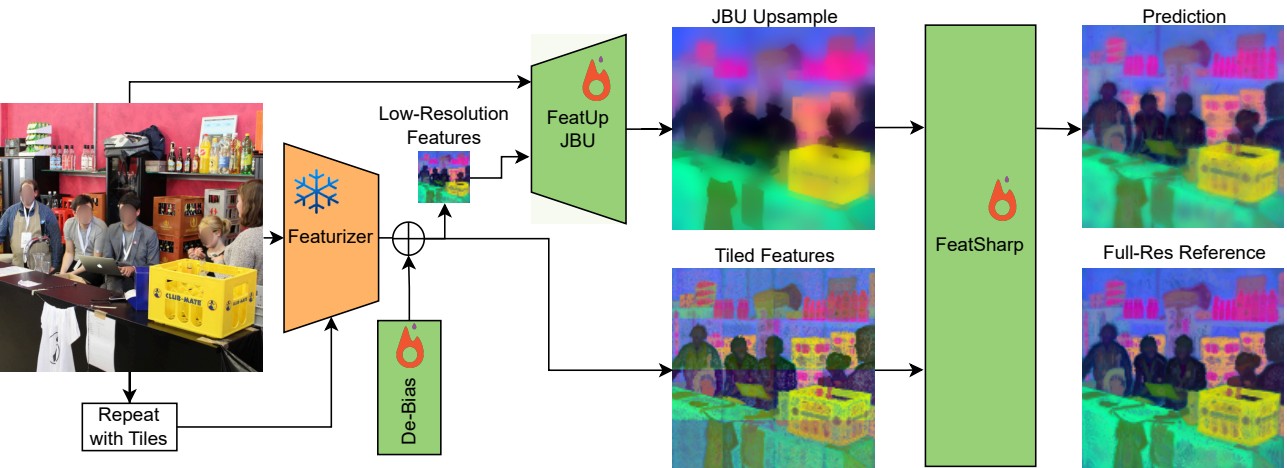

*Figure 2.* Upsampling architecture diagram. We combine the upsampled features coming from FeatUp (Fu et al., 2024) with the tiled features and mix them with FeatSharp to produce a feature map with higher fidelity. The tiled features have more detail, but also representation issues such as the difference in upper and lower body at the tile boundary. "Full-Res Reference" is for display purposes, as for a model that doesn't exhibit stable resolution scaling (e.g. DFN CLIP, SigLIP, etc.) we don't have access to a target hi-res feature map. Learned modules have a fire icon, and frozen modules a snowflake.

left the use of VFMs in a tricky spot: Transformers are computationally demanding and have poor algorithmic scaling properties ($O(n^2)$ for 1D sequences, or $O((w \cdot h)^2$ for 2D inputs), leaving the majority of models to be relatively low-resolution. For example, perhaps the most popular family of VFMs to date, CLIP (Radford et al., 2021), typically runs at 224 or 336px input resolutions, and produces spatial features at a 14x downsample (e.g. $224^2 \rightarrow 16^2$). Owing to the nature of learned position embeddings, ViTs also tend to be relatively inflexible to changes of input resolution, allowing for changes, but requiring finetuning (Dosovitskiy et al., 2021).

It is possible that the strict dependence on the training resolution is an artifact of the algorithm used for training, as DINOv2 (Oquab et al., 2023; Darcet et al., 2023) is quite robust to interpolating its position embeddings, producing stable features at various resolutions (Ranzinger et al., 2024b), ignoring for the moment that DINOv2, being a transformer, is expensive to use at high-resolution. A recent technique called AM-RADIO (Ranzinger et al., 2024a), borrowing ideas from ViTDet (Li et al., 2022), FlexiViT (Beyer et al., 2023), and RO-ViT (Kim et al., 2023), has attempted to create a resolution-flexible ViT, however it is still dependent on low-resolution ViTs as it distills from other seminal VFMs which are low-res only: DFN CLIP (Fang et al., 2023) and SigLIP (Zhai et al., 2023).

Recently, FeatUp (Fu et al., 2024) aims to directly address the problem of low-resolution vision features by using one of two learned upsampling algorithms: A model-specific generalized upsampler using Joint Bilateral Upsampling (JBU) (Kopf et al., 2007), or a model-specific-

image-specific implicit network. While they demonstrate particularly compelling results with their implicit network, their results using the stack of JBU filters lack refined details (shown in figure 17 in the appendix). Along with the lack of granular refinement, it's impossible for this approach to capture fine-grained details that are too small for the vision backbone to detect at its native resolution. To this end, we take inspiration from both FeatUp's JBU approach, as well as the recent trend in VLMs such as LLaVA 1.6 (Liu et al., 2024), InternVL-1.5 (Chen et al., 2024), NVLM (Dai et al., 2024b) and Eagle (Shi et al., 2024b) to tile an image, aggregating local features from a fixed-low-resolution model, to build an upsampler that simultaneously leverages the raw pixel guidance, low-res feature guidance, and regional tile guidance, resulting in substantially more detailed feature maps which are also capable of capturing details too small for the original resolution. Specifically, we:

- Build on top of FeatUp's JBU algorithm (Fu et al., 2024) by adding de-biasing and tiling fusion modules to incorporate detailed tile features, resulting in significantly higher levels of detail, with extensive experiments demonstrating effectiveness.

- Study the relationship between FeatUp's feature consistency and ViT-Denoiser's (Yang et al., 2024a) approach to cleaning the features of a ViT at its native resolution.

- Introduce an improved training setting for AM-RADIO (Ranzinger et al., 2024a) demonstrating a +0.6% improvement across the entire benchmark suite, and better teacher adapter features.

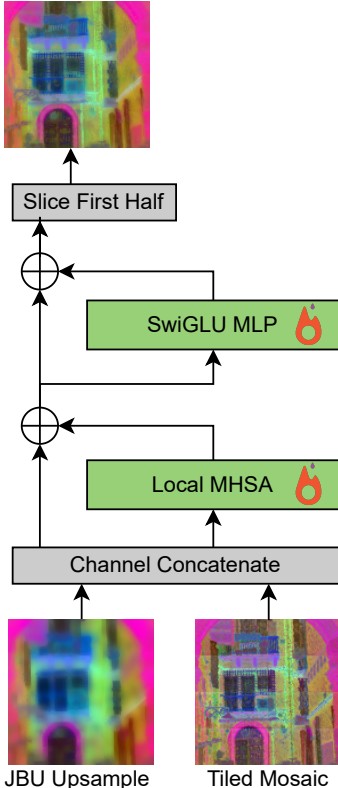

*Figure 3.* Diagram of the FeatSharp module. We first concatenate the JBU upsampled and tiled mosaic feature maps along the channel dimension. We then apply a transformer block with sliding window attention followed by MLP (in this case, SwiGLU), and then slice off the first half of the channels, corresponding to the bilinear upsampled buffer. The role of FeatSharp thus is to refine the JBU buffer by leveraging the tile buffer.

## 2. Related Work

**Feature Upsampling**   The most obvious baseline for feature upsampling is to use traditional filtering approaches such as bilinear or bicubic upsampling. The alternative is to evaluate the network at higher resolution, however it comes with the dual drawback that computational cost increases (quadratically in the case of Vision Transformers), and also that many models (ViTs in particular) have trouble extrapolating from their trained resolution (Beyer et al., 2023; Dehghani et al., 2023). If we expand our view to include parametric approaches, then deconvolution (Noh et al., 2015; Shi et al., 2016; Dumoulin & Visin, 2016) and resize-conv (Odena et al., 2016) are popular choices. There are also pixel-adaptive approaches such as CARAFE (Wang et al., 2019), SAPA (Lu et al., 2022), and FeatUp (Fu et al., 2024).

We adopt FeatUp's formulation of multi-view consistency as a way to train an upsampler, however, we notice that instead of solely relying on raw RGB pixels as guidance for upsam-

pling, we can also use a small, fixed budget of inferences (similar in spirit to their implicit model), and use a mosaic of tiles as guidance at the higher resolution. This choice gives us a richer, and semantically relevant, feature space to merge from. Additionally, it allows us to incorporate information from regions that were too small for the low-res view, but become visible within a tile. Small details are a limitation of every approach that doesn't acquire extra samples from the base model, as they rely on all relevant information already being encoded by the initial model evaluation.

**Feature Denoising**   Related to multi-view consistency, ViT-Denoiser (Yang et al., 2024a) noticed that ViT features are generally very noisy (although some are much cleaner than others), and also propose a multi-view consistency formulation to learn how to separate fixed noise, conditional noise, and semantic content. We notice the deep ties between ViT-Denoiser and FeatUp, in that multi-view consistency provides a way to eradicate fixed-pattern noise from the feature buffer. Drawing inspiration from this, we add a learnable bias buffer (similar to learned position embeddings) at the output of the base model. This simple change works because fixed patterns will degrade multi-view consistency, as the pattern is always local to the view, and lacks global coherence.

**VLMs**   The use of tiling to increase information is currently very prominent in VLMs (Liu et al., 2024; Chen et al., 2024; Dai et al., 2024a), albeit an alternative approach is to instead leverage the models at hi-res themselves (Beyer et al., 2024; Wang et al., 2024). We also see RADIOv2.5(Heinrich et al., 2024) being primarily useful at high-resolution within VLMs. In the increasingly VLM-centric approach to computer vision, we turn our focus to RADIOv2.5, as it has a training procedure that relies on matching a high-resolution student against a low-resolution teacher, an application area that is perfect for studying feature upsampling, as it would provide richer guidance to the distillation.

**Agglomerative Models**   In the agglomerative model space, there are currently three major approaches: RADIO (Ranzinger et al., 2024b;a; Heinrich et al., 2024), Theia (Shang et al., 2024), and UNIC (Sariyildiz et al., 2024). We focus our attention on RADIO because it is the only approach that directly tries to tackle resolution flexibility as well as high-resolution.

## 3. Method

We leverage FeatUp's training algorithm of treating the upsampling problem as that of multi-view consistency between the upsampled and then downsampled features and different low-res views of the same image.

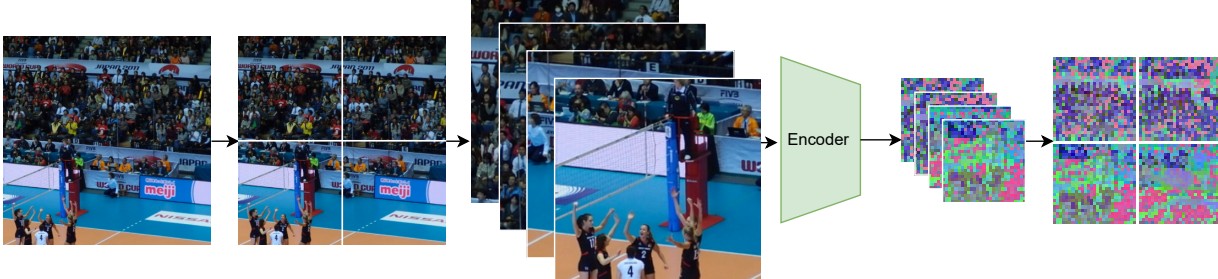

*Figure 4.* Visualization of the tiling process. An input image (left) is split into $2 \times 2$ tiles, each of which is resized to match the input resolution of the encoder, fed through the encoder independently, and then stitched back into a higher resolution feature map. Feature maps shown are from DFN CLIP, and they are resized to be larger than actual for demonstration purposes.

### 3.1. Review - FeatUp: Joint Bilateral Upsampling (JBU)

Given a high-resolution signal $G$ (e.g. the raw pixels) as guidance, and a low-resolution signal $F_{lr}$ that we'd like to upsample, and let $\Omega$ be a neighborhood of each pixel in the guidance. Let $k(\cdot, \cdot)$ be a similarity kernel that measures how close two vectors are. Then

$$
\hat{F}_{hr}[i,j] = \frac{1}{Z} \sum_{(a,b) \in \Omega} \Big( F_{lr}[a,b] \cdot \\
k_{range}\left(G[i,j], G[a,b]\right) \cdot \\
k_{spatial}\left([i,j],[a,b]\right) \Big)
\tag{1}
$$

with $Z$ being a normalization to make the kernel sum to 1. $k_{spatial}$ is a Gaussian kernel with learnable $\sigma_{spatial}$ defined as

$$
k_{spatial}(x,y) = \exp\left( \frac{-\|x-y\|_2^2}{2\sigma_{spatial}^2} \right)
\tag{2}
$$

and $k_{range}$ as

$$
k_{range}(x,y) = \operatorname*{softmax}_{(a,b) \in \Omega} \left( \frac{1}{\sigma_{range}^2} h(G[x,y]) \cdot h(G[a,b]) \right)
\tag{3}
$$

with $h(x)$ being a learned MLP projector. They define

$$
F_{hr} = (\text{JBU}(\cdot, x) \circ \text{JBU}(\cdot, x) \circ ...) (f(x), x)
\tag{4}
$$

as a stack of $2\times$ upsamplers, thus enabling power-of-2 upsample factors. With $x$ being the original input image and $f(x)$ being the low-resolution feature map. We note that $2\times$ is not a necessary part of the architecture, and that their implementation supported arbitrary factors, so we simply propose to take a given upsample factor $z \in \mathbb{Z}_+$ and prime

factorize $z$ to get a set of upsample factors, using a $\text{JBU}_k$ for each prime factor. This decomposes to an identical operation as before when $\log_2 z \in \mathbb{Z}_+$, but allows for an easy guide for any other integer, e.g. for a $14\times$ upsample corresponding to a patch-size-14 backbone, we'd use a $(\text{JBU}_{7\times} \circ \text{JBU}_{2\times})(f(x), x)$ stack.

As is typical with bilateral upsampling, this method is very sensitive to strong edges in the guidance buffer, however, it also tends to over-smooth features in regions of lower contrast. Particularly, it struggles with feature patterns such as SAM (figure 1) where there are interior edges in feature space, but not pixel space. This results in the features being blurred inside of objects.

We don't make any changes to their downsampler, instead opting to just use their Attention Downsampler without modification. We then focus on two changes, one to output normalization, and the other to how upsampling guidance is computed.

### 3.2. Feature Normalization

FeatUp supports either leaving the features coming from the backbone as-is (e.g. no normalization), or using a LayerNorm to better condition the outputs for feature learning. For a similar motivation as PHI-S (Ranzinger et al., 2024a), we want to avoid using the raw features as they have very distinct distributions, and we'd also like to avoid using LayerNorm as it makes the features incompatible with the original feature space. Naïvely learning the raw feature space across the suite of teachers without normalization often led to convergence issues, particularly given the wide variance of activations. Thus, we adopt PHI-S as a way to standardize the backbone features without distortion and to retain the ability to model the original distribution. We compute the distribution statistics over 100k samples from the training dataset.

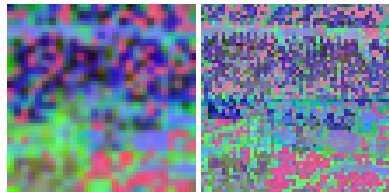

*Figure 5.* Visualization of $2\times$ upsampling using bilinear (*left*) versus tiling (*right*), using the DFN CLIP encoder.

### 3.3. Tile-Guided Attentional Refinement

Joint-Bilateral Upsampling is able to retain object boundaries primarily in instances when there are noticeable changes in intensity in the RGB input image. This results in sharp contours, but within a region, we end up with vague and blurry feature representations. Owing to the reliance on raw pixel intensities, object contours that are less discriminative in color space often get blurred with the neighborhood. Finally, because the upsampling operation is only truly operating on the low resolution feature maps of the model, it's impossible for JBU to introduce new details into the feature map that are visible/encodable at higher input resolutions. FeatUp's implicit upsampler doesn't have this same limitation because it's constructing a global view from numerous local views of the original image, enabling detailed refinements. We propose an intermediary method between JBU which leverages a single featurizer inference, and the implicit model, which relies on numerous inferences and is thus cost prohibitive[1].

Inspired by the use of tiling in Vision-Language Models (VLMs) (Liu et al., 2024; Shi et al., 2024a; Dai et al., 2024b), we develop an attentional refinement block that is able to integrate the information between a JBU upsampled feature map, as well as a feature map composed of tiles. We show an overview of the algorithm in figures 2, 3 and 4. The diagram shows actual results using RADIOv2.5-L, which is the most scale equivariant foundation model (Heinrich et al., 2024), and generally the strongest visual foundation model (Lu et al., 2024; Drozdova et al., 2024; Guo et al., 2024). Because the model has strong resolution scaling, it provides us with a good way to compare the results of the upsampling process against the feature maps of the same resolution attained by increasing the resolution of the input image. We also observe that even just at $4\times$ tiling, there are major discontinuities in the tiled feature map, which the FeatSharp module must overcome to produce a unified higher-resolution image.

For the FeatSharp module, we leverage a single Attention+SwiGLU transformer block. In order to prevent the quadratic cost of global attention, we instead use 2D local attention (Ramachandran et al., 2019). We concatenate the

JBU upsampled buffer with the tiled feature map and feed it to the block. After the block is computed, we slice the first $C$ dimensions of the output, with $C$ being the model feature dimension, and treat that as the refined features. The slicing strategy takes advantage of the fact that a transformer block has a residual pathway, and thus a no-op from the transformer would be equivalent to returning the bilinear upsampled features. Through the attention mechanism, the model is able to consider the local neighborhood and refine its features to achieve better multi-view consistency. To this end, we train our model identically to FeatUp's multi-view consistency algorithm. We do not employ any special loss functions beyond the MSE loss on multi-view consistency, contrary to FeatUp's use of Total Variation and Conditional Random Field losses. We provide ablations wrt architecture choice in appendix B.2.

### 3.4. Denoising

Drawing inspiration from (Yang et al., 2024a), we notice that the problem formulation has a very similar solution to FeatUp (and ours), owing to the fact that all methods are using multi-view consistency and thus learn to eliminate position-based artifacts. From their formulation:

$$\text{ViT}(x) = f(x) + g(\mathbf{E}_{pos}) + h(x, \mathbf{E}_{pos})$$
$$\text{((Yang et al., 2024a), Eq 5)}$$

We add a learnable $g$ buffer, such that

$$\hat{f}(x) = f(x) + g \qquad (5)$$

with $f(x)$ being the frozen vision encoder. The learnable $g$ allows our model to learn and negate the fixed position artifacts that the encoder produces. Notably, given that we are also using the base model for the tiles, this learned buffer is applied to all of the generated tiles as well. We visualize these biases in figure 11. It's entirely possible for FeatSharp to remove the biases itself, but we found that having this learnable bias buffer consistently improves multi-view consistency, which we show in table 7 in the appendix.

### 3.5. Complexity

An important point about this method is that because of the tiling, it requires more evaluations of the base vision model to construct the high-resolution feature map. However, due to the scaling properties of global self-attention, our proposed method always has better scaling properties than running the original model at higher resolution (assuming the model is capable of doing this in the first place). Specifically, let $f(x)$ be the relative cost of computing Feat-Sharp, and $g(x)$ the relative cost of running the base model

---

[1] https://github.com/mhamilton723/FeatUp/issues/2

on the hi-res input, with $x \in \mathbb{Z}_+$ being the number of tiles per dimension, and $c$ being the cost of processing a single tile:

$$
\begin{aligned}
f(x) &\le c \sum_{i=1}^{x} i^2 \\
g(x) &= c \left(x^2\right)^2 = cx^4 \\
f(x) &\le g(x) \quad \forall x > 1
\end{aligned}
\tag{6}
$$

We show the empirical scaling cost in figure 14 in the appendix, and prove equation 6 in appendix E.2. We also note that experiments for FeatSharp only use the global view, plus the final level of tiles, thus $f(x)$ simplifies to $f(x) = c \left(1 + x^2\right)$, however we prove the general case, as progressive upsampling may be beneficial in future work.

## 4. Upsampling Results

We consider upsampling to be important in cases where one is given a fixed pretrained model, and the goal is to extract more information out of it, for a given image. We study our method in relation to FeatUp from a core multi-view consistency standpoint in this section, from a semantic segmentation linear probe standpoint, and also for training a new RADIO-like model with hi-res teacher targets.

### 4.1. Fidelity

**Multi-View Consistency**  Following (Ranzinger et al., 2024a), we use their definition of fidelity (equation 51) for multi-view consistency, where a higher fidelity value means that the upsampled-transformed-downsampled representations are closer to the raw transformed predictions from the model.

$$
f(\mathbf{X}, \mathbf{Y}) = \frac{\mathrm{MSE}(\mathbf{Y}, \boldsymbol{\mu}_{\mathbf{Y}})}{\mathrm{MSE}(\mathbf{X}, \mathbf{Y})}
\tag{7}
$$

with $\mathbf{X}$ being the warped predictions and $\mathbf{Y}$ the targets. This serves as a proxy measure for how well the upsampler is working, as arbitrarily warping and downsampling it results in representations closer to the real prediction at low resolution. We show these results in figure 6, where we observe that FeatSharp consistently achieves the highest fidelities, substantially so with the "cleaner" models such as DINOv2-L, RADIOv2.5-L, and SAM-H.

### 4.2. Qualitative

We run this upsampling method on seven different foundation models coming from diverse domains such as supervised (ViT, (Dosovitskiy et al., 2021)), contrastive

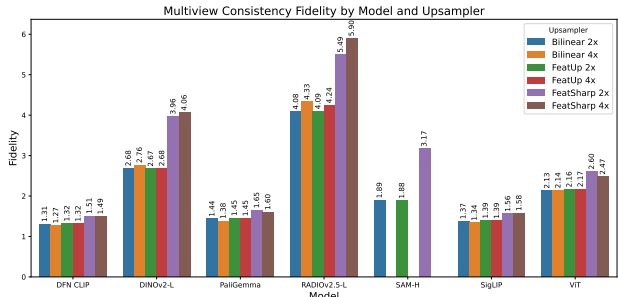

*Figure 6.* Fidelity plot for different models and upsampling methods. Higher values are better. We don't show SAM 4x because of OOM issues training these models.

(DFN CLIP (Fang et al., 2023), SigLIP (Zhai et al., 2023)), Self-supervised (DINOv2-L-reg (Darcet et al., 2023)), Segmentation (SAM (Kirillov et al., 2023)), VLM (PaliGemma (Beyer et al., 2024)), and Agglomerative (RADIOv2.5-L (Ranzinger et al., 2024a)). Results are in figure 1. The original feature maps run the spectrum from extremely noisy (SigLIP) to very clean (RADIOv2.5-L, SAM), which allows us to demonstrate the effectiveness of the approach on a diverse set of models. Taking SAM for an example, the way in which it has thick edge outlines cannot be reproduced in the shape interior by FeatUp, primarily because the bilateral upsampler is operating on the raw pixels, where the interior edge doesn't exist in the real image. For all of the featurizers, FeatSharp is able to achieve more legible representations. In particular it is more able to closely match the real hi-res features in the second column.

### 4.3. Semantic Segmentation

Semantic segmentation has the potential to benefit from increased resolution, as it allows for label contours to be more precise, and potentially for regions to be recovered that are otherwise too small. The first setting we evaluate on is we train both FeatUp and FeatSharp at $2\times$ and $4\times$ upsampling, both using PHI-S. We resize the input size to be the featurizer's native input resolution, which we call "$1\times$ Input Size", and we also consider "$2\times$ Input Size", where we double the input size, and feed directly to the featurizer in the case of "Baseline", or we allow the upsampler to have higher resolution guidance while keeping the featurizer input fixed at $1\times$ resolution. We show these results in figure 7. In most cases, both upsampling algorithms produce higher quality segmentations than the baseline, however, FeatUp is worse than the "Baseline $2\times$" method for RADIOv2.5-L and ViT. In all cases, FeatSharp is superior to both FeatUp and also the baselines by significant margins. We even improve upon SOTA RADIO's published result of 51.47 with a $2\times$ upsampling combined with $2\times$ input size, producing a model that attains 53.13 mIoU, a $+1.66$ mIoU improvement. RADIO itself improves with the $2\times$ input size, but not to the same degree as with FeatSharp, with FeatSharp being $57\%$ faster.

We notice that $3\times$ upsampling is generally slightly worse than $2\times$ or $4\times$ for both upsamplers, but leave an investigation into why as future work. This figure also provides insight into the general ability of these foundation models to operate at resolutions that deviate from their native resolution. The CLIP family models (DFN CLIP, SigLIP, PaliGemma) are unable to benefit from this increased resolution at all, or in the case of PaliGemma, degrade with it, while the first-class-dense models like DINOv2 and RADIO natively benefit from increased resolution. Surprisingly, even though ViT is solely trained as a classification model, it also benefits from native resolution increases.

### 4.4. Object Detection

We integrate our method, FeatUp-JBU, the baselines, as well as SAPA (Lu et al., 2022) and the preprint ReSFU (Zhou et al., 2025) into Detectron2 using the Edge[2] codebase, and probe on COCO 2017 (Lin et al., 2014). We use a [(*frozen*) featurizer] + [(*frozen*) upsampler] + ViT-Det (Li et al., 2022) + DINO (Zhang et al., 2023)[3] (DETR with Improved DeNoising Anchor Boxes for End-to-End Object Detection) pipeline. We evaluate these methods on both RADIOv2.5-L (Heinrich et al., 2024) and the recently proposed SigLIP2-SO400M-512 (Tschannen et al., 2025) models. We show the results in table 1, where FeatSharp is clearly best able to improve object detection results over baseline and comparison methods, particularly for small objects, benefitting from the additional tile guidance. We also note that we were unable to use SAPA with SigLIP2 due to a CUDA configuration error in their backprop kernel.

### 4.5. Agglomerative Models

We build upon RADIOv2.5-L (Heinrich et al., 2024) as it learns directly from the spatial features of teacher models. In particular, we consider whether we can improve upon their multi-resolution training strategy by using FeatSharp to convert the low-res teachers into hi-res teachers. We convert the teachers in the bottom left quadrant "Low-Res Teacher / High-Res Student" in their Figure 6 into "High-Res Teacher / High-Res Student" by using the upsampler. We consider a few different comparative baselines in order to prove the efficacy of the technique. For our baseline, we make one change to the recipe in (Heinrich et al., 2024), which is to bilinearly upsample the teacher to match the student, as opposed to downsampling the student. We stress that this produces a strong baseline, as it scores even better than RADIOv2.5-L on average. The reason we make this change is so that we're across the board comparing upsampling methods, with bilinear being the simplest technique. Then,

---

[2]https://dgcnz.github.io/edge/part2/adapting.html

[3]Not to be confused with the DINO/DINOv2 foundation models.

| Upsampler | Upsample Factor | AP | | | |
|---|---|---|---|---|---|
| | | * | Sm | Md | Lg |
| **RADIOv2.5-L** | | | | | |
| Baseline | 1 | 51.38 | 28.73 | 56.56 | 73.72 |
| Bilinear | 2 | 51.61 | 28.43 | 56.98 | 74.14 |
| SAPA | 2 | 41.44 | 15.92 | 45.08 | 69.77 |
| ReSFU | 2 | 49.81 | 26.22 | 55.37 | 73.55 |
| FeatUp | 2 | 46.71 | 21.77 | 52.01 | 72.25 |
| FeatSharp | 2 | **54.83** | **34.72** | **59.40** | **74.40** |
| **SigLIP2-SO400M-512** | | | | | |
| Baseline | 1 | 52.66 | 30.31 | 57.94 | 74.31 |
| Bilinear | 2 | 52.69 | 30.19 | 57.84 | 74.16 |
| SAPA[†] | 2 | - | - | - | - |
| ReSFU | 2 | 50.84 | 28.45 | 56.18 | 73.69 |
| FeatUp | 2 | 47.42 | 22.87 | 53.17 | 72.80 |
| FeatSharp | 2 | **55.93** | **36.85** | **61.00** | **74.62** |

*Table 1.* COCO 2017 object detection results using Detectron2 and various upsampling methods for both RADIOv2.5-L and SigLIP2-SO400M. [†]SAPA was unable to process this model's input size/dimension, producing a CUDA configuration error.

we consider two techniques which are popular in the VLM literature: Tiling (Liu et al., 2024), and S2 (Shi et al., 2024a). Both of these rely on tiling, but S2 also considers the low-res version. Because we need the feature space to remain the same as the low-res partition of RADIO, we opt to upsample the low-res feature map, and then interpolate the upsampled-low-res against the tiled version, using $y = \beta \cdot \text{low-res} + (1 - \beta) \cdot \text{high-res}$. We set $\beta = 0.5$ as it's unclear what an optimal balance might be, and it's expensive to search this space. As a final baseline, we include FeatUp's JBU variant, as the implicit version would be prohibitive to use within a training loop[4].

In figure 8 we qualitatively visualize the DFN CLIP adaptor features learned by the radio model. We can see that each upsampling method has a substantial impact on the resulting feature maps. The baseline method exhibits strong high-frequency artifacting starting at 768px. This is likely when RADIO "mode switches" to high-resolution, which is something that (Heinrich et al., 2024) addressed for the backbone features, but apparently still exhibit for the adaptor features. We observe that Tiling and S2 exhibit not only high-frequency noise patterns like the baseline, but also obvious grid patterns, arising from the use of tiles. More troublesome, we can see how the student learned to mimic representation switches within tiles for both Tiling and S2, where the mulch in one cell gets a different feature representation (thus color) than another, based on whether any of the dog is present in the tile view. FeatUp appears to mode switch starting at 768px into a smooth, but low-detail feature space. FeatSharp remains smooth and highly de-

---

[4]1-5 minutes per image

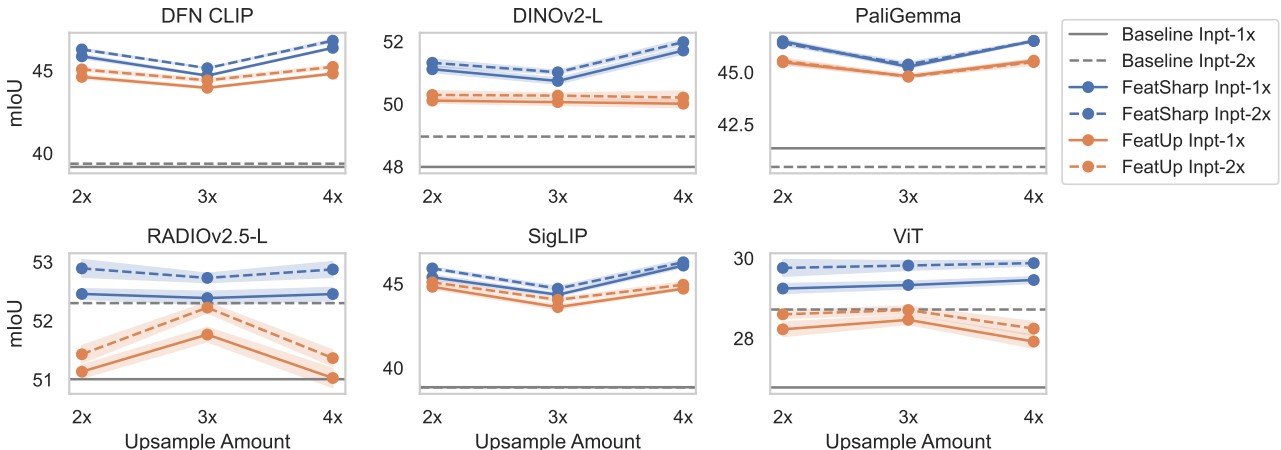

*Figure 7.* ADE20k (Zhou et al., 2017) Semantic segmentation results for different featurizers and upsamplers. We also vary the input size between Inpt-1× and Inpt-2× the featurizer's native resolution. 1× Resolutions: DFN CLIP = 378px, DINOv2-L = 448px, PaliGemma = 448px, RADIOv2.5-L = 512px, SigLIP = 378px, ViT = 224px. The dark line represents the mean of 5 runs, with shaded areas showing the standard deviation. Because the x-axis is the upsample amount, the baselines should technically be single points on a "1x" x-coord, but we instead draw a line to make it easier to see the change in the upsamplers across the upsample amounts. E.g. for "RADIO, Baseline Inpt-2x", we can see that it's better than FeatUp 2× upsampling, but worse than FeatSharp 2× upsampling.

tailed as resolution increases, however, visually, it's still possible that the features are mode switching. We show another comparison in appendix F with the SigLIP adaptor head.

Along with improvements in the adaptors, we also study the effects on the backbone features for the RADIO model. Following (Maninis et al., 2019; Lu et al., 2024) we report the MTL Gain ($\Delta_m$) across a suite of tasks. Unlike the prior works, instead of leveraging a single-task baseline, we opt to report the change relative to the baseline training run.

Let

$$\delta_m = 100 \cdot (-1)^{l_t} \frac{M_t - M_{B,t}}{M_{B,t}} \qquad (8)$$

$$\Delta_m = \frac{1}{T} \sum_{t=1}^{T} \delta_m \qquad (9)$$

where $M_t$ is the metric for the current model on task $t$, and $M_{B,t}$ is the metric for the baseline model. $l_t$ is 0 when higher task values are better, and 1 when lower is better.

We show the MTL Gain results in table 2. Given that the results are relative to our baseline run, S2 and FeatSharp are the only two methods to improve, however, only FeatSharp was categorically better, leading to a +0.39% improvement across all benchmarks on average. These two methods are the only two that incorporate both low-res and hi-res features, with S2 perhaps being considered a baseline to Feat-Sharp, so their improvements suggest that this extra detail is indeed useful for RADIO training. We also see that our

version of RADIO with FeatSharp teachers generally does better than RADIOv2.5-L (Heinrich et al., 2024), which is the current state of the art, where we improve over it on everything except for the VILA task. We report all of the raw per-task benchmark scores in tables 3, 4, 5 and 6 in the appendix.

## 5. Conclusion

We have presented a novel feature upsampling technique named FeatSharp that achieves higher multi-view fidelity than the current best method, FeatUp. We achieve this by joining FeatUp's JBU upsampler with a mosaic of tiles, and then process with a single local attention block. We demonstrate its effectiveness on ADE20K semantic segmentation linear probing, where the use of FeatSharp improves over both baseline and FeatUp, even with the strongest segmenter, RADIO, which itself can handle hi-res inputs robustly. We also demonstrate our effectiveness in object detection with frozen backbone and upsampler, and see AP benefits in particular for small objects, but also medium and large. We then demonstrate the effectiveness of FeatSharp by employing it directly within RADIO training, enabling hi-res distillation targets for low-res-only teacher models. In doing so, our FeatSharp-RADIO largely improves on dense vision task benchmarks, and yields an overall improvement over our reproduction baseline, which itself improves over RADIOv2.5-L, the current state of the art. We believe this work can be useful both as a drop-in extension of existing vision systems which rely on pretrained vision encoders, as well as the newly trained FeatSharp-RADIO model with hi-res teachers, which can emulate these same models. Ow-

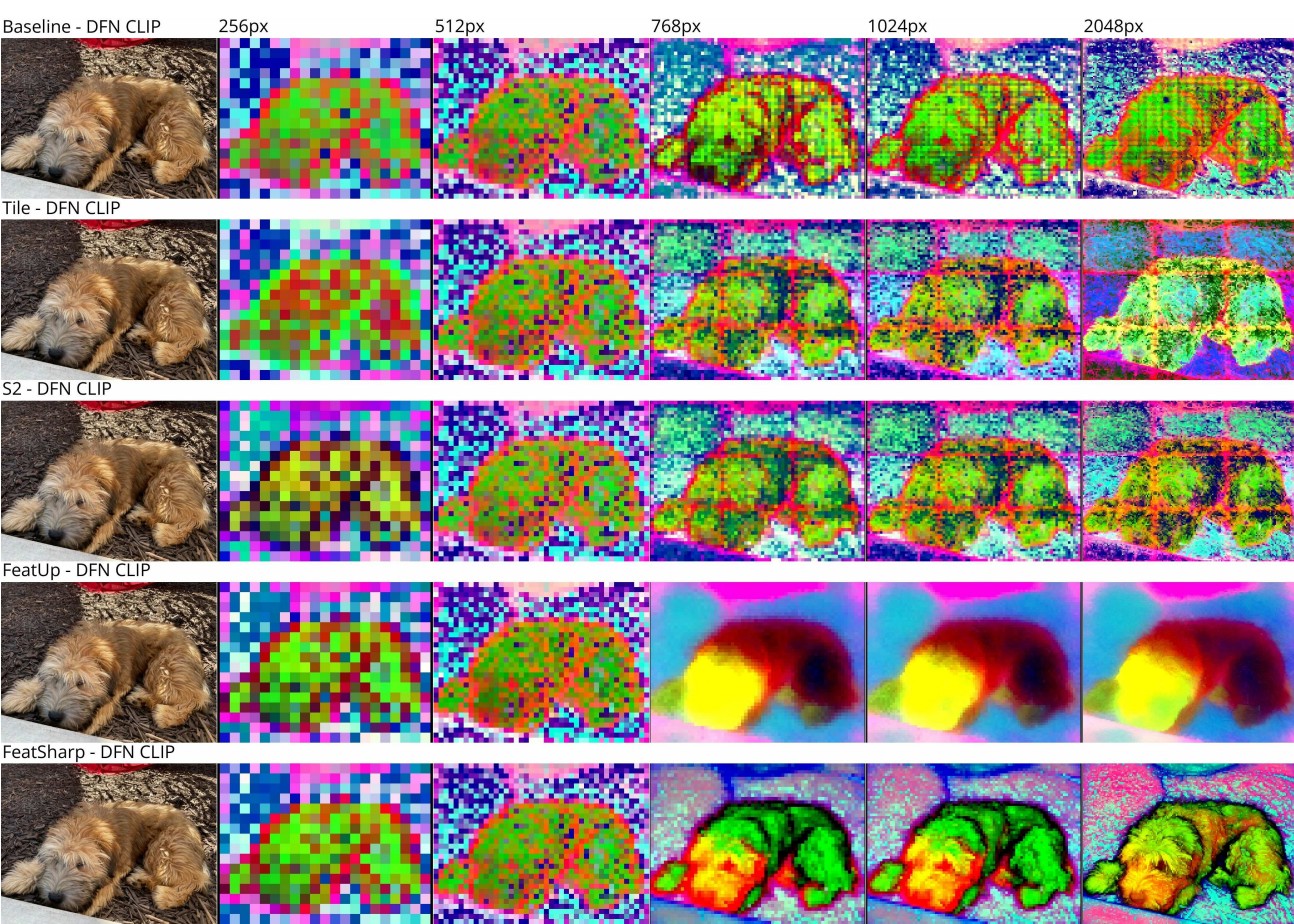

*Figure 8.* Visualization of our trained RADIO's DFN CLIP adaptor when the high-res partition used various teacher upsample schemes.

| Upsampler | Classification | Dense | Probe 3D | Retrieval | Pascal Context | NYUDv2 | VILA | $\Delta_m\%$ |
|---|---|---|---|---|---|---|---|---|
| RADIOv2.5-L | -0.47 | -0.09 | -1.05 | -0.45 | **0.62** | -2.26 | **2.24** | -0.21 |
| Baseline | 0.00 | 0.00 | 0.00 | 0.00 | 0.00 | 0.00 | 0.00 | 0.00 |
| Tile | -0.03 | **0.30** | -0.08 | -0.23 | -0.02 | **1.33** | -3.17 | -0.27 |
| S2 | -0.05 | 0.15 | -0.03 | -0.44 | 0.13 | **1.33** | -0.89 | 0.03 |
| FeatUp | -0.07 | 0.14 | 0.23 | -0.07 | 0.14 | 0.32 | -1.58 | -0.13 |
| FeatSharp | **0.06** | 0.16 | **0.83** | **0.13** | 0.17 | 0.93 | 0.43 | **0.39** |

*Table 2.* Relative changes (in %) on a suite of aggregated benchmarks, with each column reporting $\delta_m\%$ and averaged into $\Delta_m\%$. All relative changes are against our baseline run. Raw metrics are in section A.1. *NOTE: The upsamplers are only applied to the DFN CLIP and SigLIP teachers during RADIO training. Metrics are collected from trained RADIO without upsampling methods.*

ing to FeatSharp-RADIO's emulation abilities, it allows us to estimate these teacher models at arbitrary resolutions, not just integer upsampling factors as restricted in Feat-Sharp/FeatUp's core training algorithm. Further, combining RADIO's "ViTDet" (Li et al., 2022) mode with these hi-res teacher emulations allows us to achieve hi-res feature maps without fully paying the quadratic penalty in number of tokens as required by standard ViTs.

## Impact Statement

This paper presents work whose goal is to advance the field of computer vision. By virtue of being a lightweight addition to existing vision models, the work aims to open up doors for higher-resolution perception tasks (e.g. segmentation, depth perception, distillation, etc.) while retaining the original model representations. As such, the ethical impacts are constrained to those of the model being upsampled. The FeatSharp training code will be released to the community.

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

# A. RADIO Results

## A.1. Benchmark Results

In this section, we provide detailed benchmark results used to compute the MTL aggregate metrics in table 2. We show these results in tables 3, 4, 5, and 6.

| Upsampler | Classification | | Zero Shot Retrieval | | | |
|---|---|---|---|---|---|---|
| | ImageNet-1k | | COCO | | Flickr30k | |
| | Zero Shot | kNN | Text2Im | Im2Text | Text2Im | Im2Text |
| RADIOv2.5-L | 81.01 | 84.68 | 51.65 | **69.06** | 77.52 | 90.80 |
| Baseline | 81.47 | 85.00 | **52.25** | 68.68 | **78.64** | 90.60 |
| Tile | 81.41 | **85.01** | 51.90 | 68.30 | 78.46 | 91.10 |
| S2 | 81.44 | 84.95 | 51.94 | 67.98 | 78.34 | 90.80 |
| FeatUp | 81.39 | 84.96 | 51.93 | 68.40 | 78.26 | **91.70** |
| FeatSharp | **81.56** | **85.01** | 52.13 | 68.80 | 78.50 | 91.30 |

*Table 3.* Classification and Zero Shot Retrieval Metrics. All zero shot methods use the DFN CLIP Text encoder, paired with RADIO's respective learned adaptor.

| Upsampler | Dense | | | Probe3d | | | |
|---|---|---|---|---|---|---|---|
| | ADE20k | VOC | SAM COCO | Depth | Surface Normals | Correspondence | SPair71k |
| RADIOv2.5-L | 51.47* | 85.49* | 75.06 | 84.69 | 60.06 | 58.46 | 54.36 |
| Baseline | 51.58 | 85.08 | 75.46 | 85.03 | **61.42** | 59.27 | 54.49 |
| Tile | 51.62 | **85.55** | **75.67** | 85.14 | 60.85 | 59.65 | 54.41 |
| S2 | 51.56 | 85.28 | 75.66 | 85.11 | 60.49 | **59.84** | 54.68 |
| FeatUp | 51.67 | 85.20 | 74.54 | 85.39 | 61.20 | 59.63 | 54.63 |
| FeatSharp | **51.75** | 85.13 | 75.54 | **85.48** | 60.76 | 59.55 | **56.33** |

*Table 4.* Dense and Probe3D (El Banani et al., 2024) metrics. *We report numbers for evaluation at 512px, which are found in Table A5 in RADIOv2.5 (Heinrich et al., 2024).

| Upsampler | Pascal Context | | | | NYUDv2 | | |
|---|---|---|---|---|---|---|---|
| | SemSeg mIOU ↑ | Parsing mIoU ↑ | Saliency maxF ↑ | Surface Normals ↓ | SemSeg mIoU ↑ | Depth rmse ↓ | Surface Normals ↓ |
| RADIOv2.5-L | 82.87 | 74.32 | **81.65** | **16.15** | 61.42 | 0.458 | 18.57 |
| Baseline | 82.88 | 75.02 | 80.55 | 16.49 | 62.64 | 0.448 | 18.09 |
| Tile | 83.07 | 75.28 | 80.56 | 16.60 | **62.91** | 0.437 | 17.90 |
| S2 | 83.09 | **75.45** | 80.63 | 16.56 | 62.64 | **0.436** | **17.86** |
| FeatUp | 83.11 | 75.21 | 80.68 | 16.51 | 62.74 | 0.449 | 17.93 |
| FeatSharp | **83.17** | 75.28 | 80.64 | 16.51 | 62.60 | 0.439 | 17.95 |

*Table 5.* Pascal Context and NYUDv2 multitask learning metrics. Following the setup of MLoRE (Yang et al., 2024b) and RADIOv2.5 (Heinrich et al., 2024) with a convolutional probe. NOTE: We're only using their harness with a conv probe, and not using their architecture.

## A.2. Additional Qualitative Visualizations

In figure 9, we show more PCA feature visualizations coming from our trained RADIO models. We can see that RADIO learned to mimic how tiling lacks global context, as the background-only tiles use a different feature space than those with background+content.

## A.3. Difference Visualization

In figure 10, we show the difference heatmaps between FeatSharp/FeatUp and Bilinear upsampling. For DFN CLIP and SigLIP, we actually see that a lot of the differences are with high frequency noise. More intuitively, for the cleaner RADIO and SAM models, the differences are largely concentrated at the edges. Because the PCA projection down to 3D can sometimes distract from the true differences between representations (e.g. color flipping), these difference maps help show where the information is truly different between methods.

| Upsampler | AI2D No Mask Accuracy | ChartQA Overall | DocVQA Val Accuracy | GQA Accuracy | InfoVQA Val | MME Perception | MMMU Val | OCR Bench Accuracy | POPE F1 | SEED All | TextVQA Val |
|---|---|---|---|---|---|---|---|---|---|---|---|
| RADIOv2.5-L | **79.2** | 56.4 | **49.2** | 63.4 | **29.8** | **1592.4** | **43.3** | **441** | 87.6 | **69.27** | **66.7** |
| Baseline | 78.04 | 57.32 | 47.12 | 63.41 | 28.78 | 1568.11 | 40.00 | 422 | 87.51 | 69.08 | 65.33 |
| Tile | 75.71 | 54.32 | 42.44 | 63.60 | 26.80 | 1541.61 | 40.33 | 400 | 86.63 | 68.62 | 63.78 |
| S2 | 77.07 | 55.28 | 44.89 | 63.73 | 28.75 | 1549.50 | 42.33 | 405 | 87.14 | 68.96 | 64.86 |
| FeatUp | 78.40 | 55.56 | 45.31 | 63.60 | 26.98 | 1563.57 | 40.33 | 407 | 86.83 | 68.57 | 65.05 |
| FeatSharp | **79.15** | **57.56** | 46.39 | **63.75** | 28.25 | 1564.41 | 42.22 | 416 | **88.06** | 68.77 | 66.41 |

*Table 6.* VILA metrics, using the same setup from [(Heinrich et al., 2024), Table 9].

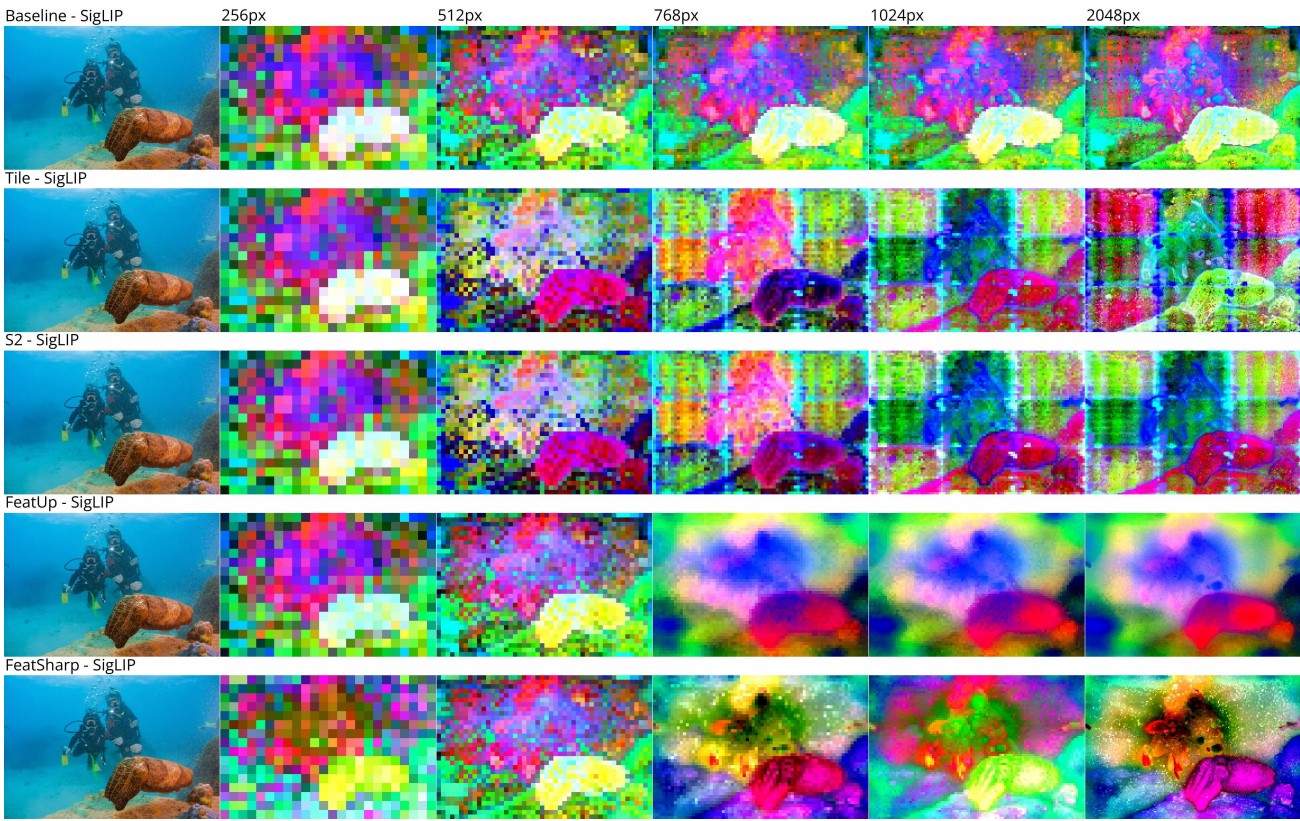

*Figure 9.* Visualization of RADIO's SigLIP adaptor, using different teacher upsampling techniques.

## B. Architecture Ablations

### B.1. De-bias Module

Adding the de-bias module yields a positive improvement in fidelity across all featurizers studied. We show the changes in fidelity metrics for all featurizers in table 7. We also demonstrate that this module helps for both FeatSharp and FeatUp, as it occurs prior to upsampling, and is thus generally applicable. In figure 11, we visualize the learned biases, which are unique to each featurizer, but also how these biases can sometimes be directly visible in the output features of these models. Most obvious is SAM, which has windowing artifacts stemming from their use of windowed attention.

### B.2. FeatSharp Architecture

**Input Feature Selection** Based on figures 2 and 3, there are important degrees of freedom in the design of the system. We demonstrate in table 7 that including the de-bias module always improves the distribution matching fidelity. Here, we look at some of the other design choices:

- Should we use bilinear upsampling, or FeatUp, for the low-res upsampler? (or both)

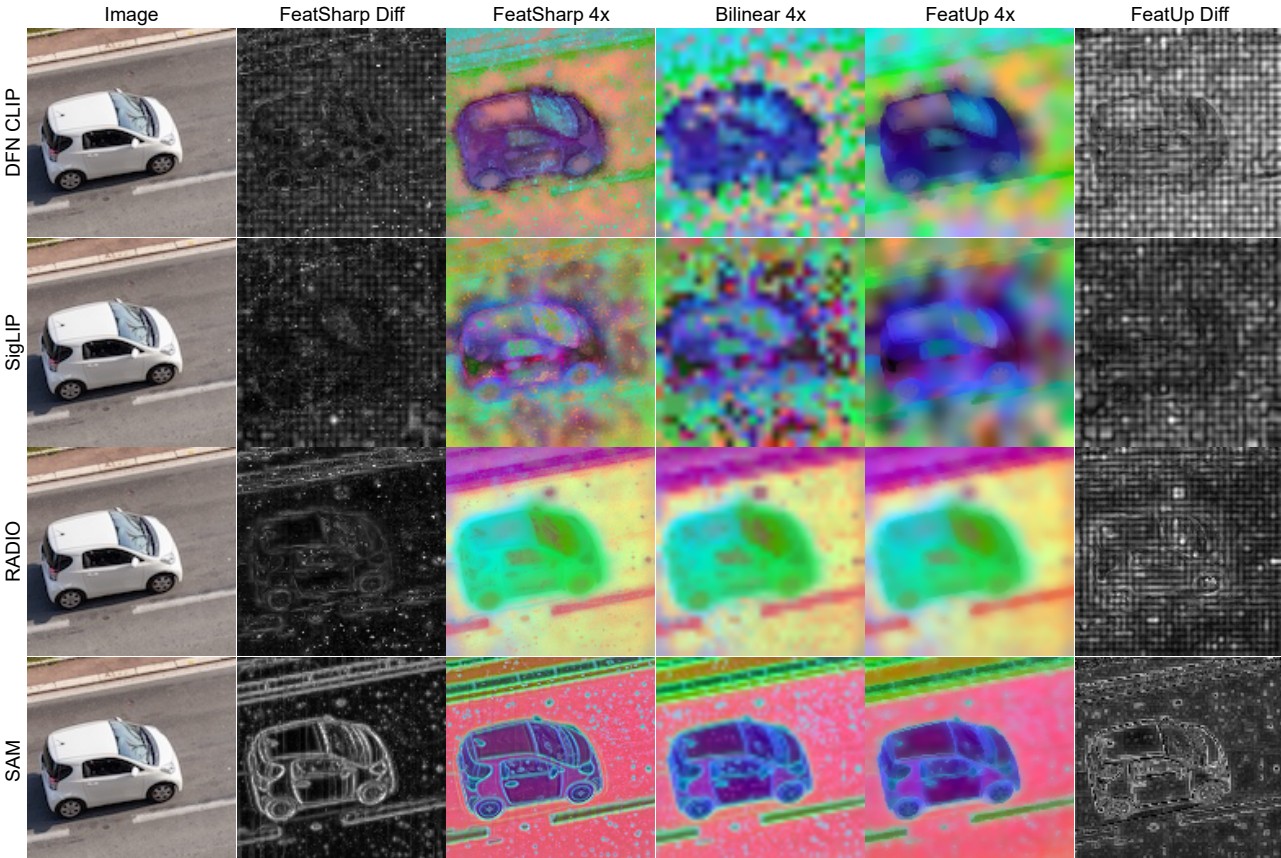

*Figure 10.* Visualization of the differences between the FeatSharp or FeatUp algorithm, and bilinear upsampling.

- Should bilinear, tiling, or FeatUp be the residual pathway to the output?

- Should we use all three upsampling methods?

We show the results of this ablation in table 8 for both our noisiest featurizer (SigLIP), and our cleanest (RADIO) as a sanity check that we aren't overfitting to a particular featurizer. We also note that we're relatively agnostic to the specific base feature upsampler, allowing us to use other methods, such as ReSFU (Zhou et al., 2025), as future work.

We also visualize the resulting feature maps of the different input configurations in figure 12, as it's hard to get a feel for what this multi-view fidelity is telling us. It's clear both in the metrics (table 8) and the visualization that just using one of the three different feature maps largely retains the biases of those views (e.g. the bilinear result is roughly regular bilinear upsampling, the FeatUp input looks like vanilla FeatUp, etc.). We can also see the profound impact on the resulting maps based on which input feature map is the residual pathway. For the single input case, we can observe how the tile-only input results in a distribution shift, apparent because the color space has largely shifted. Once we look at 2+ inputs, the color spaces become consistent. Even though the bilinear-first configurations always have the highest fidelity, they also are clearly the blurriest. This is perhaps not surprising given that FeatUp's JBU upsampler has a strong edge prior, so incorporating it into FeatSharp will also hone in on edge boundaries. Also, regardless of 2+ input configuration, we can see that FeatSharp is able to refine the text, clearly leveraging the tile features. The similarity is very close to the tile-only input in that region. We do notice that using "Bilinear + *other(s)*" yields the highest fidelities, but also that the resulting feature maps are relatively blurry.

In order to not make an entire argument to prefer the use of FeatUp's JBU as the low-res upsampler due to the prettiness of the PCA features, we also consider alternative measures of the produced features. The Total Variation (TV) loss gives us a sense of how much "noise" is present in the produced features, simply based on accumulating the differences between neighbors. On its own, this doesn't tell us much, but in conjunction with the multi-view-consistency fidelity, and when that

| Model | FeatSharp 2x | FeatSharp 4x | FeatUp 2x | FeatUp 4x |
|---|---|---|---|---|
| DFN CLIP | 0.020 | 0.022 | 0.033 | 0.025 |
| DINOv2-L | 0.121 | 0.110 | 0.164 | 0.144 |
| PaliGemma | 0.017 | 0.021 | 0.030 | 0.023 |
| RADIOv2.5-L | 0.208 | 0.173 | 0.144 | 0.138 |
| SAM-H | 0.067 | | 0.076 | |
| SigLIP | 0.014 | 0.017 | 0.033 | 0.019 |
| ViT | 0.014 | 0.009 | 0.096 | 0.038 |

*Table 7.* The delta change in multi-view consistency fidelity when applying the learned de-bias buffer. Positive values mean that the fidelity has improved, which is true for every model and upsampler tested.

| Arch | SigLIP | | | RADIO | | |
|---|---|---|---|---|---|---|
| | Fidelity ↑ | TV Loss ↓ | CRF Loss ↓ | Fidelity ↑ | TV Loss ↓ | CRF Loss ↓ |
| **Single Input** | | | | | | |
| Bilinear | 1.466 | 0.135 | 0.137 | 4.599 | 0.061 | 0.071 |
| FeatUp | 1.440 | **0.020** | **0.051** | 3.870 | **0.025** | **0.047** |
| Tiles | 1.261 | 0.897 | 0.237 | 2.713 | 0.357 | 0.094 |
| **Two Inputs** | | | | | | |
| Bilinear + Tiles | 1.470 | 0.136 | 0.135 | **4.694** | 0.073 | 0.074 |
| FeatUp + Tiles | 1.460 | 0.072 | 0.062 | 4.173 | 0.076 | 0.057 |
| Tiles + Bilinear | 1.337 | 0.812 | 0.241 | 3.202 | 0.336 | 0.098 |
| Tiles + FeatUp | 1.323 | 0.821 | 0.228 | 3.157 | 0.337 | 0.094 |
| **Three Inputs** | | | | | | |
| Bilinear First | 1.469 | 0.138 | 0.137 | 4.682 | 0.072 | 0.073 |
| FeatUp First | **1.473** | 0.063 | 0.065 | 4.202 | 0.064 | 0.057 |
| Tiles First | 1.339 | 0.800 | 0.238 | 3.238 | 0.334 | 0.098 |

*Table 8.* Ablation over different FeatSharp-2x configurations. Single Input means that we only supply the respective buffer to the FeatSharp module. For "Two Inputs", we compare different low-res upsamplers in conjunction with tiling, and also the residual pathway, where the first value indicates the residual path. The FeatSharp module must integrate the information from the other value into the residual. "Three Inputs" is similar to Two, except that we only care about which buffer is the residual path, owing to the fact that there's no intrinsic order preference in the weights for the secondary buffer(s). "TV Loss" stands for Total Variation Loss (Rudin et al., 1992). CRF is Conditional Random Field, and is essentially measuring how similar the semantics of two nearby RGB pixel patches are based on how similar the RGB values are. TV and CRF losses were not included in the gradient during training.

fidelity is roughly equal, it might be reasonable to assume that less variation is better. We can see in table 8 that the use of FeatUp does indeed reduce this for SigLIP, but has the opposite effect on RADIO. The other prior that we consider is the CRF loss, which approximately translates to the idea that nearby regions that have a similar RGB color should probably also have similar semantics. The JBU also does a good job of reducing this for our noisiest SigLIP model, as well as for RADIO. It stands to reason that spurious model noise is penalized by CRF because it breaks visual/semantic correspondence. For both TV and CRF losses, we capture the metrics, but they do not participate in the gradient. So, we're purely measuring the latent behaviors.

An alternative argument, which doesn't require hand waving about whether less variance is a good thing, or if spatio-semantic similarity is necessarily good, we turn to Maximum Mean Discrepancy (MMD, (Gretton et al., 2012)) which is precisely defined as a way to test whether two sets of observations $X := \{x_1, ..., x_m\}$ and $Y := \{y_1, ..., y_n\}$ are sampled from the same distribution. It has the clear advantage in our setup in that $m$ doesn't have to be equal to $n$, or rather, we can have a different number of samples in $X$ than that in $Y$. Because we're upsampling, if we let the low-res distribution be $X$, then the high-res distribution can be $Y$, and then $n = u^2 m$ with $u$ being the upsampling factor. Given a radial basis function kernel (RBF)

$$k(x, y) = e^{-\gamma \|\mathbf{x} - \mathbf{y}\|^2} \tag{10}$$

then we have

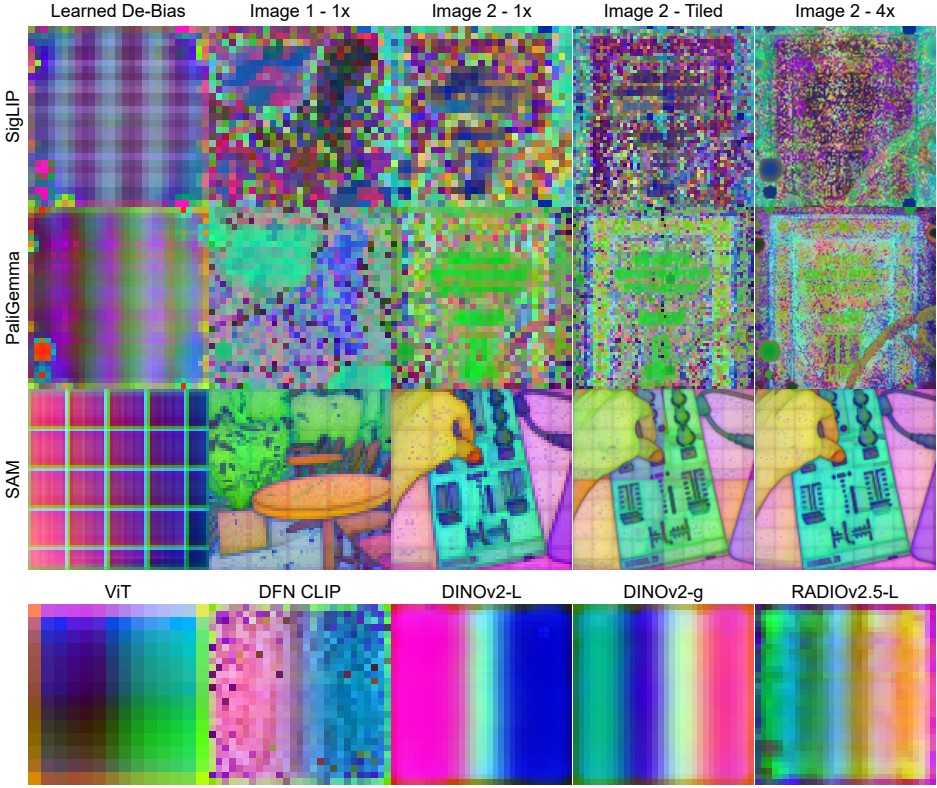

*Figure 11.* Visualization of the learned position biases for different models. All models have a bias signature, however some have very noticeable artifacts, which we visualize for SigLIP, PaliGemma, and SAM, where it's possible to see the artifacts in multiple different images and scales. We display the biases of the less apparent models in the bottom row.

$$\text{MMD}_u^2[X, Y] = \frac{1}{m(m-1)} \sum_{\substack{i,j \in m \\ i \neq j}}^{m} k(x_i, x_j) + \frac{1}{n(n-1)} \sum_{\substack{i,j \in n \\ i \neq j}}^{n} k(y_i, y_j) - \frac{2}{mn} \sum_{i}^{m} \sum_{j}^{n} k(x_i, y_j)$$

((Gretton et al., 2012), Eq 3)

we select $\gamma = \text{med}(\|x_i - x_j\|^2) \quad i \neq j$. We then collect results for Fidelity, TV Loss, CRF Loss, and MMD, for $4\times$ upsampling, and display the results in table 9. We collect these results for SigLIP, DFN CLIP, and RADIO. It is clear that FeatSharp achieves the highest upsampling fidelities across the board. FeatUp produces the lowest TV and CRF losses. It achieving the lowest TV loss is intuitive given how smooth it tends to make object interiors, seen in the pca visualizations. We can see that the lower TV and CRF losses extends to FeatSharp when we apply JBU upsampling, as it achieves lower values than using bilinear upsampling for the residual pathway. The "JBU + Tiles" FeatSharp variant also does better on MMD versus "Bilinear + Tiles" across the board. It's curious that JBU alone has the worst MMD (probably due to over-smoothing), but the best when incorporated into FeatSharp (probably owing to smoothing out the noise). We can also see that generally either "X + Tiles" FeatSharp method produces similar fidelities, aside from RADIO, where bilinear actually does do a bit better. Most likely, this is because RADIO features are themselves already fairly clean, and at some point, the structural priors of JBU actually hurt, because they're eliminating some of the raw signal that bilinear upsampling preserves. In this case, the model always has access to the raw low-res signal with bilinear upsampling because we use an integer multiple upsampling factor, and our local attention window size is larger than this multiple. Given the totality of evidence, we choose to select "JBU + Tiles" as the default upsampling mechanism, as it's either the best, or nearly so, across the board, and particularly, it does better with the vision models that are not able to natively change their resolution very well. We also note that newer methods, as they emerge, could serve as better core upsampler modules than bilinear/JBU, and can be trivially swapped in.

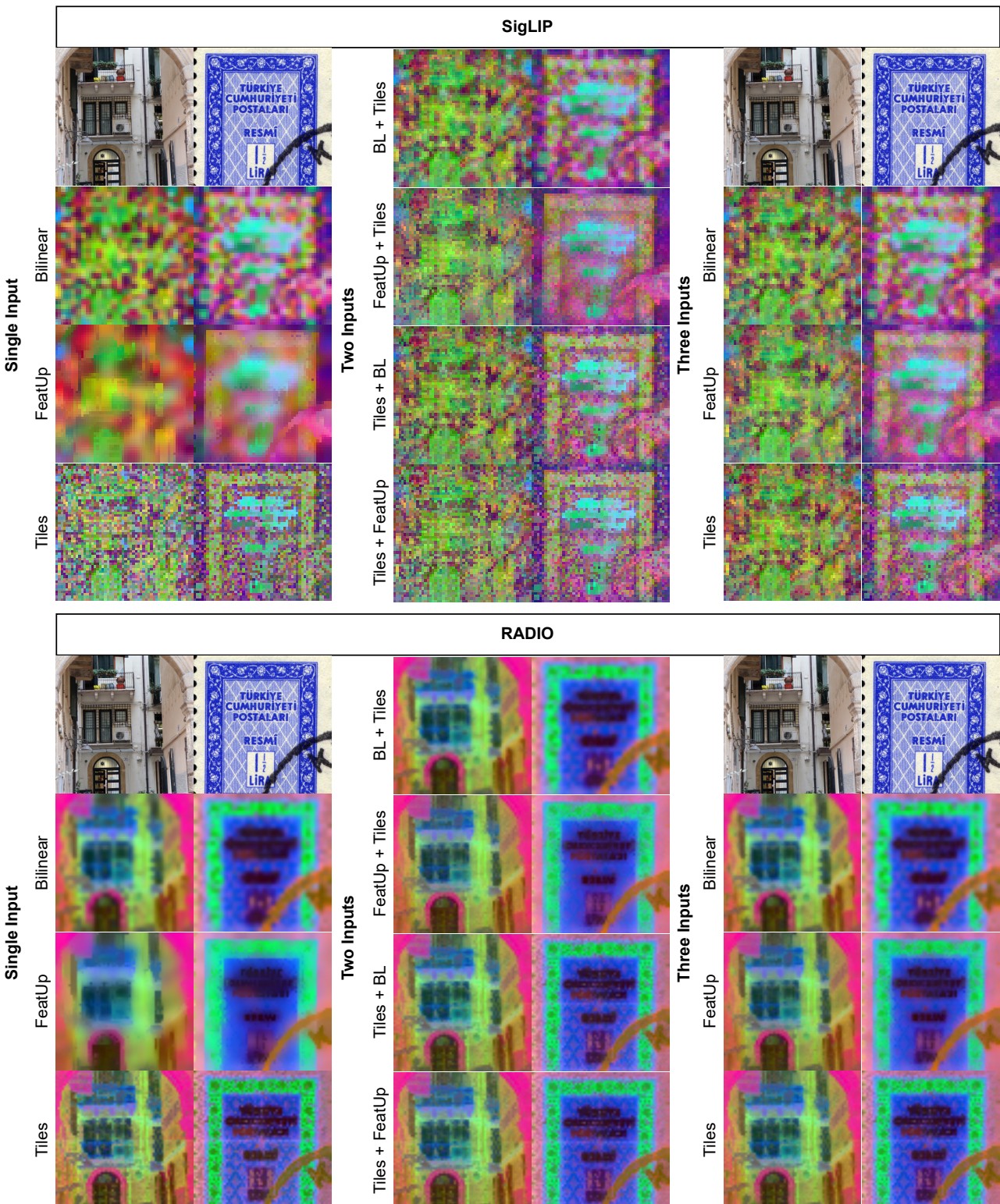

*Figure 12.* Feature visualizations of different input configurations for 2x upsampling.

| Upsampler | Featurizer (4× Upsample, Long Recipe) | | | | | | | | | | | |
| | SigLIP | | | | DFN CLIP | | | | RADIO | | | |
| | Fidelity ↑ | TV ↓ | CRF ↓ | MMD ↓ | Fidelity ↑ | TV ↓ | CRF ↓ | MMD ↓ | Fidelity ↑ | TV ↓ | CRF ↓ | MMD ↓ |
|---|---|---|---|---|---|---|---|---|---|---|---|---|
| Bilinear | 1.348 | 0.048 | 0.129 | **0.016** | 1.284 | 0.051 | 0.088 | 0.015 | 3.796 | 0.023 | 0.071 | 0.003 |
| FeatUp (JBU) | 1.375 | **0.009** | **0.047** | 0.025 | 1.326 | **0.012** | **0.032** | 0.023 | 3.680 | **0.015** | **0.064** | 0.003 |
| Bilinear + Tiles | 1.522 | 0.105 | 0.093 | 0.020 | 1.484 | 0.167 | 0.062 | 0.014 | **5.921** | 0.112 | 0.073 | 0.001 |
| JBU + Tiles | **1.580** | 0.103 | 0.077 | 0.017 | **1.493** | 0.157 | 0.046 | **0.013** | 5.898 | 0.095 | 0.065 | **0.001** |

*Table 9.* Metrics for 4× upsampling across SigLIP, DFN CLIP, and RADIO. We primarily compare whether to use bilinear or JBU upsampling for the residual branch of the FeatSharp module, but also report the same values for our two baseline methods, bilinear upsampling itself, and FeatUp (aka JBU upsampling).

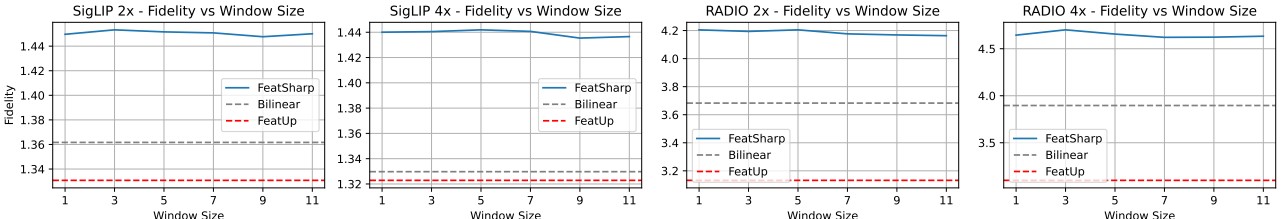

*Figure 13.* Ablation study over the choice of window size and upsampling factor for the FeatSharp module.

**Local Attention Window Size** In figure 13 we run an ablation over local attention window sizes between 1 and 11. We notice that either 3 or 5 appear to be optimal.

**Do We Even Need Attention/MLP?** As can be seen in figure 13, the choice of window size has a very small impact on the resulting fidelity. However, we can also see that FeatSharp is achieving much higher fidelity scores than Bilinear and FeatUp. So, we also study what effect the attention block, and the MLP, are having on the resulting quality. We use the "Bilinear + Tile" input configuration from section B.2, and when applicable, use a window size of 5 from section B.2. We show these results in table 10. We notice that it's not until the long recipe where at inclusion of attention is helpful, and that goes a long way toward explaining the relative insensitivity to the window size in figure 13. Inspired by figure 12, we notice that the longer training recipe results in much sharper images. In table 10 we study the effect of running just the MLP for the "Long" recipe. We can see that while the fidelity continues to improve, it doesn't keep up with the "Attention + MLP" setting, demonstrating that the attention module is indeed helpful.

## C. Implementation Details

**Upsampler Training** We leverage the same training harness as in FeatUp (Fu et al., 2024), including leveraging the same attention downsampler. We disable the use of the CRF loss that was present in the FeatUp config. Parameters in table 11.

| Modules | Fidelity | | | | | | | |
| | SigLIP | | | | RADIO | | | |
| | 2× Upsample | | 4× Upsample | | 2× Upsample | | 4× Upsample | |
| | Short | Long | Short | Long | Short | Long | Short | Long |
|---|---|---|---|---|---|---|---|---|
| Linear | 1.521 | 1.581 | **1.508** | 1.566 | 4.702 | 5.397 | 4.926 | 5.701 |
| Attention | 1.505 | 1.566 | 1.500 | 1.560 | 4.410 | 5.253 | 4.656 | 5.446 |
| MLP | **1.522** | 1.581 | 1.506 | 1.566 | **4.741** | 5.397 | **4.934** | 5.707 |
| Attention + MLP | 1.513 | **1.584** | 1.502 | **1.568** | 4.668 | **5.502** | 4.849 | **5.711** |

*Table 10.* Fidelity metrics for different combinations of blocks in the FeatSharp module (3). The "Long" recipe trains for 3× longer than the short recipe. We only study the "MLP" vs "Attention + MLP" configurations in the long recipe because those were the top two configurations in the short recipe.

| Hyperparameter | FeatUp JBU | Regular | Long |
|---|---|---|---|
| Num GPUS | 1 | 8 | 8 |
| Batch Size (per GPU) | 4 | 4 | 4 |
| Batch Size (total) | 4 | 32 | 32 |
| Num Steps | 2,000 | 3,000 | 9,000 |
| Optimizer | NAdam | NAdam | NAdam |
| Learning Rate | 0.001 | 1e-4 | 1e-4 |
| Downsampler | Attention (k=7) | Attention (k=7) | Attention (k=7) |
| Num Jitters | 5 | 5 | 5 |
| CRF Weight | 0.001 | 0 | 0 |
| TV Weight | 0 | 0 | 0 |
| Feature Normalization | LayerNorm | PHI-S | PHI-S |
| Dataset | COCO | SA-1B | SA-1B |
| Multi-view Augs | Scale, Shift | Scale, Shift, HFlip, Rotate, Perspective | |

*Table 11.* Training hyperparameters. "FeatUp JBU" refers to the settings in the official https://github.com/mhamilton723/FeatUp. Unless otherwise specified, we report numbers based on the "Long" schedule, which includes FeatUp reproduction values, to maintain fairness.

**RADIO Training** We follow the staged setup in (Heinrich et al., 2024) section 4.2, with stages 1 and 2 being exactly identical. For stage 3, in the hi-res student branch, instead of bilerp downsampling the student features to match DFN CLIP and SigLIP (RADIOv2.5 baseline), we use our various upsampling methods to create hi-res feature maps which the student matches. We use our trained $3\times$ upsamplers for the task, as they're the smallest factor that produces feature maps larger than RADIO's hi-res partition. For FeatSharp, because we have the learned de-bias buffer which operates on the original model resolution, we also choose to apply this to the teachers in the low-res partition, as it represents the fixed bias of the teacher model, and is thus not particularly useful information.

## D. Additional Benchmarks

### D.1. Probe3d

In table 12 we show the result of various configurations in Probe3d's (El Banani et al., 2024) depth probing for both DFN CLIP and RADIO. We can see that FeatUp produces the best results, however, we also demonstrate that this is likely due to the strong structural prior to the method, as the single best configuration was to use a FeatUp JBU stack with randomly initialized weights. Both FeatUp and FeatSharp are able to strongly improve over any configuration of regular DFN CLIP. For RADIO, we can see that both FeatUp and FeatSharp are still able to improve over baseline, albeit the margins are much smaller. While FeatSharp $4\times$ does achieve the highest scores, the margin is too small to be significant compared to $2\times$ and FeatUp, but still better than baseline. We observe essentially the same trend in table 13, where the random JBU stack works the best for DFN CLIP, and then FeatUp/FeatSharp are comparable for RADIO.

### D.2. NYUDv2

We also report metrics on NYUDv2 (Nathan Silberman & Fergus, 2012) in table 14 for both DFN CLIP and RADIO, similar to Probe3d configurations. We use the MLoRE (Yang et al., 2024b) harness and their conv probing for all configurations. We only use features from the final layer of the models. We can see here that unlike Probe3d, FeatSharp does a noticeably better job than FeatUp across the board, and with FeatSharp $2\times$, we get the strongest results for DFN CLIP. For RADIO, it's much tighter between FeatSharp and Baseline, however, FeatSharp is significantly better than FeatUp.

## E. Throughput Analysis

### E.1. Empirical Throughput

In (6), using $f(x) = c(1 + x^2)$ (e.g. non-progressive tiling), we predict that based on the quadratic scaling of attention, theoretically FeatSharp should always be cheaper than running the base model at the upsampled resolution. FeatSharp's cost is linear in the number of tokens, whereas a ViT is quadratic. In figure 14, we show the results of this prediction on actual hardware. As can be seen with the "Actual" curve, the picture is a bit more complex than pure quadratic scaling, as

| Vision Encoder | Input Res | Upsampling Method | Output Tokens | Depth (Scale Aware) | | | | Depth (Scale Invariant) | | | |
|---|---|---|---|---|---|---|---|---|---|---|---|
| | | | | d1 ↑ | d2 ↑ | d3 ↑ | RMSE ↓ | d1 ↑ | d2 ↑ | d3 ↑ | RMSE ↓ |
| DFN CLIP | $378^2$ | - | $27^2$ | 0.303 | 0.575 | 0.772 | 0.168 | 0.440 | 0.710 | 0.842 | 0.134 |
| | $756^2$ | - | $54^2$ | 0.291 | 0.558 | 0.757 | 0.173 | 0.426 | 0.695 | 0.829 | 0.140 |
| | $1512^2$ | - | $108^2$ | 0.280 | 0.535 | 0.733 | 0.181 | 0.399 | 0.664 | 0.805 | 0.152 |
| | $378^2$ | 2× Upsample features | $54^2$ | 0.301 | 0.573 | 0.773 | 0.168 | 0.443 | 0.713 | 0.844 | 0.133 |
| | $(2 \times 2) \times 378^2$ | Tiling | $54^2$ | 0.248 | 0.489 | 0.697 | 0.193 | 0.354 | 0.616 | 0.771 | 0.165 |
| | $(4 \times 4) \times 378^2$ | Tiling | $108^2$ | 0.218 | 0.434 | 0.634 | 0.212 | 0.317 | 0.567 | 0.732 | 0.184 |
| | $378^2$ | FeatUp 2× | $54^2$ | 0.430 | 0.712 | 0.851 | 0.128 | 0.538 | 0.793 | 0.894 | 0.107 |
| | $378^2$ | FeatUp 4× | $108^2$ | 0.435 | 0.716 | 0.853 | 0.128 | 0.542 | 0.796 | 0.896 | 0.107 |
| | $378^2$ | FeatUp 4× (Random Weights) | $108^2$ | **0.440** | **0.723** | **0.858** | **0.126** | **0.554** | **0.805** | **0.900** | **0.105** |
| | $378^2$ | FeatSharp 2× | $54^2$ | 0.398 | 0.685 | 0.837 | 0.136 | 0.512 | 0.772 | 0.882 | 0.113 |
| | $378^2$ | FeatSharp 4× | $108^2$ | 0.419 | 0.705 | 0.847 | 0.131 | 0.527 | 0.785 | 0.890 | 0.109 |
| RADIO | $512^2$ | - | $32^2$ | 0.472 | 0.749 | 0.873 | 0.118 | 0.584 | 0.827 | 0.916 | 0.097 |
| | $1024^2$ | - | $64^2$ | 0.478 | 0.756 | 0.877 | 0.115 | 0.589 | 0.831 | 0.918 | 0.095 |
| | $2048^2$ | - | $128^2$ | 0.456 | 0.739 | 0.868 | 0.120 | 0.571 | 0.820 | 0.911 | 0.099 |
| | $512^2$ | FeatUp 2× | $64^2$ | 0.482 | 0.764 | 0.885 | 0.114 | 0.606 | 0.840 | 0.921 | 0.092 |
| | $512^2$ | FeatUp 4× | $128^2$ | 0.481 | 0.763 | 0.885 | 0.114 | 0.604 | 0.838 | 0.920 | 0.092 |
| | $512^2$ | FeatSharp 2× | $64^2$ | 0.480 | 0.766 | 0.887 | 0.113 | 0.604 | 0.840 | 0.923 | 0.091 |
| | $512^2$ | FeatSharp 4× | $128^2$ | **0.487** | **0.769** | **0.888** | **0.112** | **0.610** | **0.843** | **0.924** | **0.090** |

*Table 12.* Probe3D - Depth metrics. Linear probe over output features. "Random Weights" refers to a randomly initialized, untrained, model.

| Vision Encoder | Input Res | Upsampling Method | Output Tokens | Recall | | | |
|---|---|---|---|---|---|---|---|
| | | | | Avg | 0.01m | 0.02m | 0.05m |
| DFN CLIP | $378^2$ | - | $27^2$ | 49.26 | 26.02 | 44.47 | 77.30 |
| | $756^2$ | - | $54^2$ | 47.06 | 23.55 | 41.72 | 75.89 |
| | $1512^2$ | - | $108^2$ | 41.59 | 18.08 | 35.30 | 71.41 |
| | $378^2$ | FeatUp 2× | $54^2$ | 54.40 | 30.99 | 51.20 | 81.02 |
| | $378^2$ | FeatUp 4× | $108^2$ | 54.62 | 31.05 | 51.45 | 81.36 |
| | $378^2$ | FeatUp 4× (Random Weights) | $108^2$ | **55.72** | **32.23** | **53.05** | **81.89** |
| | $378^2$ | FeatSharp 2× | $54^2$ | 53.00 | 31.21 | 49.05 | 78.73 |
| | $378^2$ | FeatSharp 4× | $108^2$ | 53.62 | 31.49 | 49.60 | 79.77 |
| RADIO | $512^2$ | - | $32^2$ | 59.49 | 37.20 | 56.44 | 84.82 |
| | $1024^2$ | - | $64^2$ | 58.23 | 37.21 | 54.70 | 82.77 |
| | $2048^2$ | - | $128^2$ | 57.22 | 34.99 | 53.53 | 83.15 |
| | $512^2$ | FeatUp 2× | $64^2$ | 60.39 | 38.52 | 57.51 | 85.16 |
| | $512^2$ | FeatUp 4× | $128^2$ | **60.72** | 39.01 | 57.87 | **85.29** |
| | $512^2$ | FeatSharp 2× | $64^2$ | 60.69 | **40.11** | **57.95** | 84.02 |
| | $512^2$ | FeatSharp 4× | $128^2$ | 60.46 | 39.95 | 57.61 | 83.81 |

*Table 13.* Probe3D - NAVI Correspondence. "Random Weights" refers to a randomly initialized, untrained, model.

between 1x and 3x upsample factors, the scaling is actually sub-linear, which likely reflects the period where self-attention is memory bound, and not compute bound, thus adding extra tokens doesn't proportionally increase the cost. Specifically, at 1.85x upsampling, we achieve the lowest time per token, and from then on, the cost approximately linearly increases (note that time per token is the first derivative of the time per image, so linear growth implies quadratic scaling, as predicted). Because FeatUp only runs the featurizer once, and its upsampling operation is cheap, we can see that it achieves strong scaling regardless of resolution. FeatSharp requires $u^2 + 1$ inferences with $u$ being the upsample factor, so its cost is higher. Likely due to non-optimal kernels, we can see that FeatSharp does start operating faster than the base model until about 3.3x upsampling ($\approx 1260^2$px). However, also as predicted by (6), FeatSharp's scaling is linear.

### E.2. Proof of Equation 6

The progressive form of equation 6 is defined as $f(x) = \sum_{i=1}^{x} i^2$, and the regular form of self-attention is $g(x) = cx^4$, with $x$ being the upsampling factor per-side, and $c$ being the cost to evaluate at the base resolution. We want to show that:

$$f(x) \leq g(x) \quad \forall x > 1 \tag{11}$$

We start by rewriting the series for $f(x)$ in closed form

$$f(x) = c\frac{x(x+1)(2x+1)}{6} \tag{12}$$

| Vision Encoder | Input Res | Upsampling Method | Output Tokens | SemSeg mIoU ↑ | Depth RMSE ↓ | Surf Normals ↓ | Edge Loss ↓ |
|---|---|---|---|---|---|---|---|
| DFN CLIP | $378^2$ | - | $27^2$ | 53.15 | **0.551** | 23.49 | 0.130 |
| | $756^2$ | - | $54^2$ | 51.11 | 0.589 | 23.33 | 0.127 |
| | $378^2$ | FeatUp 2× | $54^2$ | 52.51 | 0.589 | 23.66 | 0.129 |
| | $378^2$ | FeatUp 4× | $108^2$ | 52.45 | 0.601 | 24.15 | 0.129 |
| | $378^2$ | FeatSharp 2× | $54^2$ | **54.29** | 0.579 | **23.14** | 0.126 |
| | $378^2$ | FeatSharp 4× | $108^2$ | 53.74 | 0.615 | 23.93 | **0.125** |
| RADIO | $512^2$ | - | $32^2$ | 60.80 | 0.486 | 19.45 | 0.127 |
| | $1024^2$ | - | $64^2$ | 62.15 | **0.479** | **18.55** | 0.123 |
| | $512^2$ | FeatUp 2× | $64^2$ | 60.64 | 0.490 | 19.32 | 0.124 |
| | $512^2$ | FeatUp 4× | $128^2$ | 60.55 | 0.493 | 19.57 | 0.125 |
| | $512^2$ | FeatSharp 2× | $64^2$ | **62.23** | 0.485 | 19.25 | 0.123 |
| | $512^2$ | FeatSharp 4× | $128^2$ | 61.71 | 0.511 | 19.82 | **0.122** |

*Table 14.* Multitask metrics on NYUDv2 (Nathan Silberman & Fergus, 2012) using the MLoRE (Yang et al., 2024b) convolutional probe harness.

which is the sum of squares sequence multiplied by $c$. So now

$$f(x) \leq g(x) \iff \frac{x(x+1)(2x+1)}{6} \leq x^4 \tag{13}$$

Given that $c > 0$ and that it's a constant factor on both sides, we can eliminate it.

$$2x^3 + 3x^2 + x \leq 6x^4 \tag{14}$$

and with $x > 0$, we can further simplify to

$$2x^2 + 3x + 1 \leq 6x^3 \tag{15}$$
$$6x^3 - 2x^2 - 3x - 1 \geq 0 \tag{16}$$
$$(x-1)(6x^2 + 4x + 1) \geq 0 \tag{17}$$

and thus $x - 1 \geq 0 \quad \forall x \geq 1$, and also $6x^2 + 4x + 1 > 0 \quad \forall x \in \mathbb{R}$. Therefore, $f(x) \leq g(x) \quad \forall x > 1$.

## F. Effects of "Over-Tiling"

In figure 8, we can see that RADIO had learned some idiosyncratic representations when using the Tile and S2 upsampling algorithms. The effects are also apparent in figure 9 where color spaces can entirely flip. To understand what's happening, we rely on the pretrained RADIOv2.5-L model, which has strong scale equivariance properties (Heinrich et al., 2024), and first see that as the number of tiles increases, the MSE error between the brute-force inference at a given resolution and the tiling of that resolution, increases. We show these results in figure 15. Visually, we argue that the major increases in MSE owes largely to regions that lack context, making it difficult for the encoder (in this case RADIO), to come up with a reasonable representation of the tile-crop. We visualize this in figure 16. Notably, we can see that the $8 \times 8$ tiling difference images are generally whiter, indicating a general drift towards higher error. We can also see particular tiles that have more error, such as the notecard in row 4, which gets nearly forgotten due to context. We can also see that there are a lot of errors with the car on row 5. The bottom center of the floor on row 6 has the same issue. So, while there appears to be a general upward error drift, it's exacerbated in regions without much variation.

## G. FeatUp's Two Methods

The FeatUp (Fu et al., 2024) paper presented two methods for feature upsampling: The JBU-Stack, and the Implicit network. The resulting quality of these two approaches are quite different, with the implicit network producing much finer detailed maps, but having the major drawback that it requires training a network per-image, and is thus computationally prohibitive

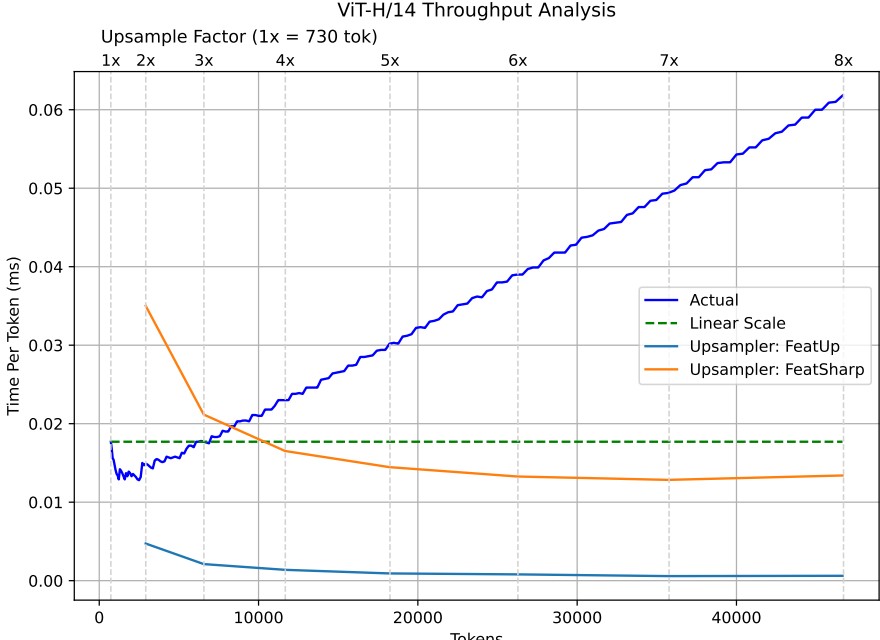

*Figure 14.* Throughput of a ViT-H/14 model (e.g. DFN CLIP) achieved with an A100 GPU, BS=1. The blue "Actual" curve reflects the time per token spent at various resolutions by the base model. "Linear Scale" assumes a constant time per token, based on the cost of 1x upsample factor. Note that "Time Per Token" is effectively the first derivative of "Time Per Image", so a linear growth in per-token represents quadratic growth in per-image.

(1̃ minute per image). The JBU stack is effective at preserving edges, but also has the effect of over-blurring object interiors. We show Figure 5 from (Fu et al., 2024) in our figure 17.

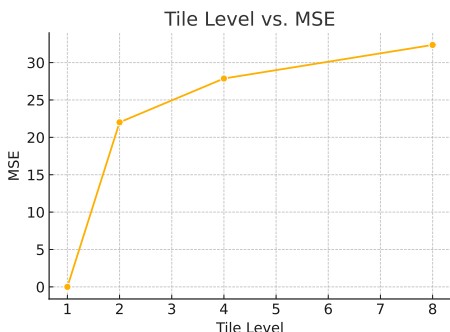

*Figure 15.* MSE error between brute-force evaluation of RADIOv2.5-L at a given resolution ($512\text{px}^2 * (\text{tile-level})$) and the tiling at the same resolution.

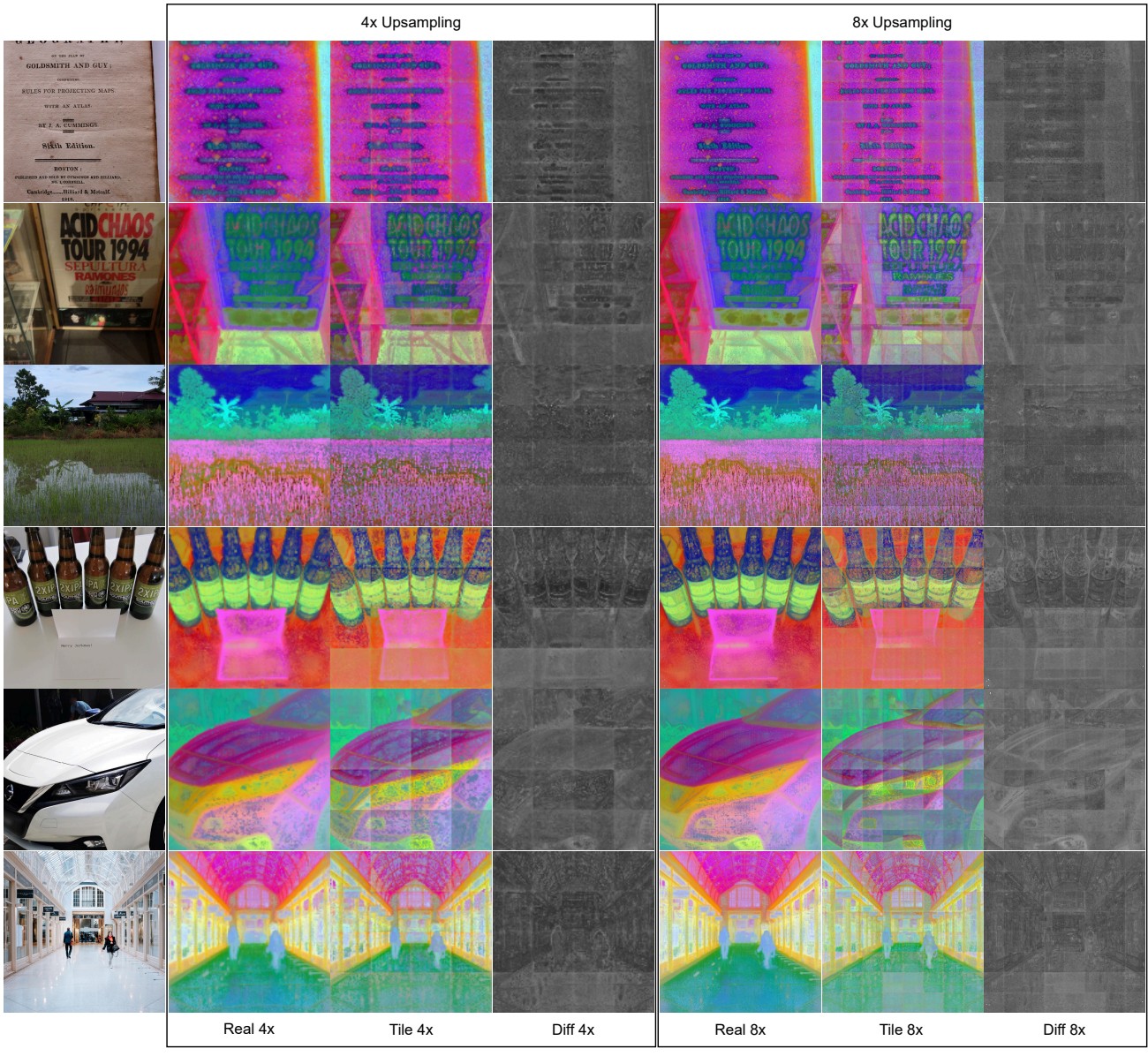

*Figure 16.* Visualization of the errors between running RADIOv2.5-L at a given resolution, and the equivalent of tiling it at the same resolution. The difference images are black when there is no difference, and white where there are large differences. The difference is computed as the euclidean distance of the full features, not their PCA projections.

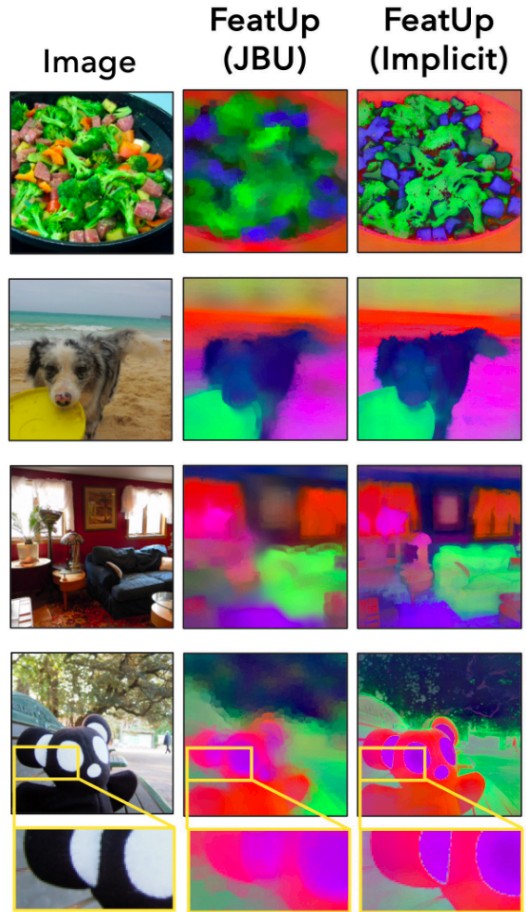

Figure 17. FeatUp's two upsampler algorithms. Taken directly from their (Fu et al., 2024) Figure 5.

