# OpenReview forum: "FeatSharp: Your Vision Model Features, Sharper"
_ICML.cc/2025/Conference — ICML 2025 poster_

### Official Review · Reviewer_Lmcr · 2025-03-09

**Overall Recommendation:** 3

**Summary:**

This paper introduces a novel upsampling method designed to address the low-resolution feature map limitations of vision encoders, particularly Vision Transformers. The proposed approach builds upon FeatUp, the current state-of-the-art upsampler, by integrating FeatUp’s Joint Bilateral Upsampling (JBU) with a mosaic of tiles, followed by processing through a single local attention block. The authors demonstrate the effectiveness of this method across various dense prediction tasks, including semantic segmentation, object detection, depth estimation, and surface normal prediction. Additionally, the study highlights how incorporating FeatSharp within RADIO training enables low-resolution-only teacher models to generate high-resolution distillation targets, further enhancing model performance.

**Claims And Evidence:**

The claims made are supported by evidence.

**Essential References Not Discussed:**

See strength and weakness.

**Ethical Review Concerns:**

No concerns.

**Experimental Designs Or Analyses:**

See strength and weakness.

**Methods And Evaluation Criteria:**

Yes the proposed methods/evaluation criteria make sense.

**Other Comments Or Suggestions:**

I recommend that the authors compare their method with the following state-of-the-art articles.

[1] Yue, Y., Das, A., Engelmann, F., Tang, S., Lenssen, J.E. (2025). Improving 2D Feature Representations by 3D-Aware Fine-Tuning. In: Leonardis, A., Ricci, E., Roth, S., Russakovsky, O., Sattler, T., Varol, G. (eds) Computer Vision – ECCV 2024. ECCV 2024.

[2] Zhou et. al. A Refreshed Similarity-based Upsampler for Direct High-Ratio Feature Upsampling. In: arXiv July 2024. https://arxiv.org/abs/2407.02283.

**Other Strengths And Weaknesses:**

Strengths:

1. Proposes a comprehensive framework for enhancing upsampling in vision encoder feature maps by effectively integrating multiple techniques, including FeatUp’s Joint Bilateral Upsampling (JBU), de-biasing, and local attention.

2. Demonstrates strong experimental results across a diverse range of vision tasks, including semantic segmentation, object detection, depth estimation, and surface normal prediction. Additionally, the study validates the effectiveness of incorporating FeatSharp into the training framework.

3. Provides a detailed cost analysis, showing that FeatSharp introduces only a minimal computational overhead compared to FeatUp, as evidenced by time-per-token evaluations.

Weaknesses:

1. Limited Novelty: The proposed method primarily builds upon FeatUp, incorporating well-established concepts such as tiling for handling high-resolution images and a simple learnable buffer for de-biasing. The integration of FeatUp, tiling, and attention layers is straightforward, and given these components, the performance improvement over FeatUp is unsurprising. The paper does not introduce fundamentally new techniques or discoveries.

2. Insufficient Baseline Comparisons: The work only compares against FeatUp, assuming it to be the sole relevant competitor for feature map upsampling. This narrow comparison overlooks other state-of-the-art methods, limiting the credibility of the results. A broader evaluation against multiple competitive approaches is necessary to justify the effectiveness of FeatSharp.

3. Limited Generalization and Marginal Gains: The paper evaluates FeatSharp exclusively within the RADIO model as the teacher, without testing its applicability to other architectures. This raises concerns about its generalizability. Moreover, the reported improvement over FeatUp is only 0.39%, which is almost negligible, calling into question the practical significance of the proposed enhancements.

**Questions For Authors:**

See Strengths and Weakness.

**Relation To Broader Scientific Literature:**

Related to vision encoder/model distillation.

**Theoretical Claims:**

No theoretical claims.

---

> ### Author Rebuttal · Authors · 2025-03-31
>
> Thank you for your thorough review.
>
> > Limited Novelty: The proposed method primarily builds upon FeatUp, incorporating well-established concepts such as tiling for handling high-resolution images and a simple learnable buffer for de-biasing. The integration of FeatUp, tiling, and attention layers is straightforward, and given these components, the performance improvement over FeatUp is unsurprising. The paper does not introduce fundamentally new techniques or discoveries.
>
> We disagree that pulling together disparate concepts that are seen in other domains of computer vision itself lends to limited novelty. Tiling is the best example of this. While it is used as a method to work around the inflexibility of most ViTs within the context of VLMs, or implicitly when doing things like sliding-window segmentation, the major insight in our work is that the tiles provide fine-grained guidance to the upsampler. This type of guidance is not present in other upsampler works, as they either rely on the raw pixels (which lack semantic meaning), or they rely on the encoder hierarchy (which isn't applicable to ViTs). ReSFU [2], for example, uses raw pixel guidance. The other critical point about upsampling has to do with the fact that small features are irrecoverable from a combination of raw-pixel + low-res featurizer. Methods like FeatUp's implicit model (>1000 views), and FeatSharp with tiling, however, are able to recover this detail because they're observing the small features. We can see this effect in Figure 13, SigLIP section, left column. These single input visualizations show what happens when you only get a single input source (plus raw pixels). We otherwise use the same model, including the single transformer block. Only the tiles observe enough detail to separate the text lines. For the RADIO section, relying on the JBU stack from FeatUp entirely misses the street lamp, blurring it into the background, and bilinear is unable to recover the latticing. Further, as part of our [rebuttal to oKYc](https://openreview.net/forum?id=lioemOcq3H&noteId=ZiLr91BWoe) we further our upsampler in an object detection setting, with additional comparisons with SAPA and ReSFU, and find that FeatSharp consistently does better. Particularly, we make the largest gains on small objects, directly providing evidence that tiles allow for the introduction of fine-grained detail. The set of techniques we chose to incorporate (multiview consistency, de-biasing, tiling, attention) are not ad-hoc, but form a cohesive novel solution to a problem.
>
> > Limited Generalization and Marginal Gains: The paper evaluates FeatSharp exclusively within the RADIO model as the teacher, without testing its applicability to other architectures. This raises concerns about its generalizability. Moreover, the reported improvement over FeatUp is only 0.39%, which is almost negligible, calling into question the practical significance of the proposed enhancements.
>
> We believe that this characterization is not supported by the evidence. We chose the RADIO training harness because it is currently the state of the art for general perception foundation models, and because it has a training setting that could directly benefit from a feature upsampler, specifically, the low-res-teacher/hi-res-student part of their training protocol. However, it's also not the case that this setting is a 0.39% improvement over FeatUp. FeatUp damages RADIO training, resulting in a -0.13% change across the benchmark suite. So FeatSharp is +0.52% relative. FeatSharp is also 0.6% better than the RADIO-AMP reported results. While 0.6% may seem small, we stress that this is for a single model over an entire suite of 31 tasks. Further, the RADIO training setting was one of two benchmark evaluations performed in the main body, the other being ADE20k semantic segmentation with six different base featurizers, and in all cases, FeatSharp was shown to be superior. In the appendix, we additionally study upsampling DFN CLIP and RADIO on Probe3d and NYUDv2 benchmarks. For an additional point of clarification, section 4.4 isn’t using an existing RADIO model as a teacher, but rather, we’re using the training protocol, and we’re upsampling DFN CLIP and SigLIP. We further chose SigLIP2-SO400M-512 as an additional featurizer in the rebuttal object detection study to provide even more evidence of broad applicability.
>
> > Insufficient Baseline Comparisons: The work only compares against FeatUp, assuming it to be the sole relevant competitor for feature map upsampling. This narrow comparison overlooks other state-of-the-art methods, limiting the credibility of the results. A broader evaluation against multiple competitive approaches is necessary to justify the effectiveness of FeatSharp.
>
> Thank you for this feedback. We have included SAPA and ReSFU in the [object detection study in our rebuttal to oKYc](https://openreview.net/forum?id=lioemOcq3H&noteId=ZiLr91BWoe) upon your advice.

---

> > ### Comment · Reviewer_Lmcr · 2025-04-02
> >
> > I appreciate the authors’ responses and the additional experiments conducted for SAPA and ReSFU. The improvements in AP small compared to other methods are noteworthy and strengthen the empirical evaluation. However, my concerns regarding the novelty of the approach and the generalization remain. Given these considerations, I am adjusting my score to borderline (weak accept).

---

### Official Review · Reviewer_8ugu · 2025-03-11

**Overall Recommendation:** 4

**Summary:**

The authors proposed a novel method for efficiently upsampling feature maps of low-resolution ViTs (CLIP) to capture fine-grained details typically lost due to limited resolution. Built upon FeatUp, their method adds de-biasing and tiles fusion modules to incorporate detailed tile features, resulting in higher levels of detail, with extensive experiments demonstrating the effectiveness.

**Claims And Evidence:**

They claim FeatSharp can upsample low-resolution feature maps while picking up on fine-grained details, and it is well evidenced by the fidelity plot in Figure 5 and other visualizations of upsampling feature comparisons.

**Essential References Not Discussed:**

No.

**Experimental Designs Or Analyses:**

Yes.

**Methods And Evaluation Criteria:**

No.

**Other Comments Or Suggestions:**

Please refer to strengths and weaknesses.

**Other Strengths And Weaknesses:**

Strengths:
1. The paper is easy to follow.
2. FeatSharp can upsample low-resolution feature maps while picking up on fine-grained details, and it is well evidenced by the fidelity plot in Figure 5.
3. The effectiveness is validated through multi-view consistency and semantic segmentation. The performance improvement on ADE20K is convincing.
4. Compared with FeatUp, the PCA visualization of FeatSharp is much closer to Real 4x.

Weaknesses:
1. When the number of tokens is larger, the inference cost of FeatSharp is much higher than FeatUp, as shown in Figure 15. Considering its performance improvement evidenced by the fidelity plot and other results, the cost is acceptable, which can be further studied as the future work.
3. The paper has some presentation mistakes. One fire sign in Fig. 1 is out of the box, and the colors of the boxes are confusing. The citation in Line 249 is confusing. One captioned data in Fig. 5 is out of the box. The presentation can be further promoted.

**Questions For Authors:**

I'm curious about the performance on fine-grained image image classification datasets, like CUB-200-2011 [1].

[1] Wah, Catherine, et al. "The caltech-ucsd birds-200-2011 dataset." (2011).

**Relation To Broader Scientific Literature:**

N/A

**Theoretical Claims:**

N/A

---

> ### Author Rebuttal · Authors · 2025-03-31
>
> We thank you for your review.
>
> > I'm curious about the performance on fine-grained image image classification datasets, like CUB-200-2011
>
> Thank you for your suggestion. Due to time and compute constraints, we were limited to running only one further study comparing upsamplers, and we selected COCO 2017 detection due to its ubiquity in the literature. We have those results in [this rebuttal to reviewer oKYc](https://openreview.net/forum?id=lioemOcq3H&noteId=ZiLr91BWoe). Given the results on COCO, it is plausible that similar effects could be observed with detailed datasets, particularly with small foreground categories, such as Caltech-UCSD Birds-200-2011, but are forced to leave the analysis to future work.
>
> > The paper has some presentation mistakes. One fire sign in Fig. 1 is out of the box, and the colors of the boxes are confusing. The citation in Line 249 is confusing. One captioned data in Fig. 5 is out of the box. The presentation can be further promoted.
>
> Thank you, we will absolutely revise these issues in a prospective camera ready.

---

> > ### Comment · Reviewer_8ugu · 2025-04-07
> >
> > The authors' reponse has addressed my major concerns, and I'll keep my rating as Accept.

---

### Official Review · Reviewer_oKYc · 2025-03-12

**Overall Recommendation:** 3

**Summary:**

The paper discusses improving vision model features by refining their sharpness and resolution. It builds on the JBU algorithm to provide more detailed feature maps and studies how to clean features effectively using ViT-Denoiser's methods. The paper also enhances the AM-RADIO framework, achieving better benchmark performance and feature adaptation.

**Claims And Evidence:**

Yes, the claims are well supported by their evidence.

**Essential References Not Discussed:**

No essential references missing, but I can recommend some papers in open-vocab segmentation that might help evaluating the FeatSharp method:

[1] Extract free dense labels from clip (ECCV'22)

[2] Clip surgery for better explainability with enhancement in open-vocabulary tasks (arxiv'23)

[3] SCLIP: Rethinking Self-Attention for Dense Vision-Language Inference (ECCV'24)

[4] Clearclip: Decomposing clip representations for dense vision-language inference (ECCV'24)

**Experimental Designs Or Analyses:**

The authors conducted experiments across a variety of visual tasks which technically sound. However, as I mentioned in “Methods And Evaluation Criteria”, some results are unsatisfactory and lack explanations and analyses. To validate the practical significance of sharp upsampling, I suggest that the authors supplement experiments on vision-language benchmarks, which involve many fine-grained reasoning tasks, such as detecting a small object in a large scene. I believe these tasks could better substantiate the claims of FeatSharp's effectiveness.

**Methods And Evaluation Criteria:**

Overall, there are two primary criteria used for evaluating the FeatSharp method.

1. The qualitative results look nice, with the upsampled feature maps produced by FeatSharp look indeed "sharper" than baseline reseults.

2. However, the quantitative results are not very convincing to me. For example, in Fig. 7, the neumerical results on semantic segmentation, why does 3x sampling generally underperform both 2x and 4x? Intuitively, if the upsample method is correct, the semantic segmentation task should continuously benefit from the high-definition feature map. However, the authors observed counterintuitive results but did not provide explanations or analyses. Additionally, in most experiments in the appendix, I also found that FeatSharp does not consistently bring quantitative gains across various visual prediction tasks. These issues raise concerns about the significance and contributions of the methods in this paper: Merely looking good in visualizations does not prove the method's effectiveness. The authors need more numerical superiority in the results to demonstrate the specific benefits of the new model for visual tasks.

**Other Comments Or Suggestions:**

There are some minor issues in the paper which might cause confusion. Eg, what does "2x upsampling" exactly mean? Does it mean scale up both width and height by 2x or scale up the area by 2x?

**Other Strengths And Weaknesses:**

See comments above.

**Questions For Authors:**

No more questions.

**Relation To Broader Scientific Literature:**

It has a close relationship to open-vocabulary semantic segmentation tasks witch typically leverage CLIP to perform pixel-level predictions. It might also be a good idea t

**Theoretical Claims:**

I did not find incorrect theoretical claims in the paper.

---

> ### Author Rebuttal · Authors · 2025-03-31
>
> Thank you for your detailed review.
>
> > what does "2x upsampling" exactly mean?
>
> We increase the width and height by 2x.
>
> > However, the quantitative results are not very convincing to me. For example, in Fig. 7, the neumerical results on semantic segmentation, why does 3x sampling generally underperform both 2x and 4x? Intuitively, if the upsample method is correct, the semantic segmentation task should continuously benefit from the high-definition feature map. However, the authors observed counterintuitive results but did not provide explanations or analyses.
>
> We agree that we're missing the analysis for the 3x case. However, it's not necessarily true that increasing the resolution of the features should increase semantic segmentation benchmarks. Our goal is to produce higher-resolution features that are consistent with the featurizer itself, and that is occasionally at odds with doing well on a particular downstream task. For example, resolution is not the reason that PaliGemma is worse than DINOv2-L which is worse than RADIOv2.5-L, because they all encode the same number of tokens. Instead, in figure 7, what we observe is that FeatUp/FeatSharp are cleaning up the noisy models by favoring the view-consistent representations (compared to the baseline horizontal lines). We further observe that FeatSharp consistently does better than FeatUp on the task. An alternative way to look at this is by comparing "Baseline Inpt-1x" against "Baseline Inpt-2x". This gives us a sense of whether it's merely increasing the resolution that yields better segmentation. We see that DINOv2-L and RADIOv2.5-L improve when increasing resolution. However, the noisy models (DFN CLIP, PaliGemma, SigLIP) either see negligible change, or even get a bit worse.
>
> We observe a similar effect in tables 11, 12, and 13 where the optimal choice of upsampler depends on the task and the featurizer itself. For DFN CLIP in the NAVI Correspondence task, it appears as though a simple bilateral upsample works the best, with entirely untuned weights, owing to both encoder noise, and also perhaps OOD alignment between the model and task. For table 13, we see that FeatSharp operates similarly to running RADIO at the resolution that matches the number of output tokens, with FeatSharp always being better than the low-res baseline (512px input).
>
> > Additionally, in most experiments in the appendix, I also found that FeatSharp does not consistently bring quantitative gains across various visual prediction tasks. These issues raise concerns about the significance and contributions of the methods in this paper: Merely looking good in visualizations does not prove the method's effectiveness. The authors need more numerical superiority in the results to demonstrate the specific benefits of the new model for visual tasks.
>
> Thank you for this feedback, as it has motivated us to run another benchmark setting in order to provide further evidence of the efficacy of our method. We have integrated our method, as well as SAPA [3], and ReSFU [4], into Detectron2, and evaluated on COCO 2017. We used Edge [5], which allowed us to create a <featurizer>+<upsampler>+DINO [6] harness. We chose to use RADIOv2.5-L and SigLIP2-SO400M-512 [7] as featurizers, with the latter to further demonstrate versatility.
>
>
> ### RADIOv2.5-L (512px input)
> |Upsampler|Upsample Factor|Fidelity|AP|AP Small|AP Medium|AP Large|
> |-|-|-|-|-|-|-|
> |Baseline|1||51.38|28.73|56.56|73.72|
> |Bilinear|2||51.61|28.43|56.98|74.14|
> |SAPA|2|2.81|41.44|15.92|45.08|69.77|
> |ReSFU|2|3.69|49.81|26.22|55.37|73.55|
> |FeatUp|2|3.71|46.71|21.77|52.01|72.25|
> |FeatSharp|2|**5.35**|**54.83**|**34.72**|**59.40**|**74.40**|
>
>
> ### SigLIP2-SO400M-512 (512px input)
> |Upsampler|Upsample Factor|Fidelity|AP|AP Small|AP Medium|AP Large|
> |-|-|-|-|-|-|-|
> |Baseline|1||52.66|30.31|57.94|74.31|
> |Bilinear|2||52.69|30.19|57.84|74.16|
> |SAPA**|2|
> |ReSFU|2|1.62|50.84|28.45|56.18|73.69|
> |FeatUp|2|1.54|47.42|22.87|53.17|72.80|
> |FeatSharp|2|**1.89**|**55.93**|**36.85**|**61.00**|**74.62**|
>
> ** *SAPA failed in backprop with a cuda configuration error, due to the size of the feature map and channel count of the model.*
>
> While FeatSharp improves the metrics across the board, it makes its largest improvements on AP Small, followed by AP Medium. This makes intuitive sense as the tiling allows for the detection of small objects that otherwise are missed by any upsampler which doesn't have access to multiple views. FeatSharp provides a holistic method which not only keeps representations consistent as resolution increases, but also allows for the incorporation of new details which are missed by the low res encoding, which is an important advancement of the literature.
>
> [3] [SAPA](https://arxiv.org/pdf/2209.12866)
>
> [4] [ReSFU](https://arxiv.org/pdf/2407.02283)
>
> [5] [Edge](https://dgcnz.github.io/edge/part2/adapting.html)
>
> [6] [DINO](https://arxiv.org/abs/2203.03605) (*not to be confused with DINOv2*)
>
> [7] [SigLIP2](https://arxiv.org/abs/2502.14786)

---

> > ### Comment · Reviewer_oKYc · 2025-04-09
> >
> > Thanks for the rebuttal. My concerns are addressed and I raise my score to 3.

---

### Official Review · Reviewer_qjbo · 2025-03-14

**Overall Recommendation:** 4

**Summary:**

The paper introduces FeatSharp, a method which builds upon FeatUp (specifically its JBU upsampling variant) [1],  by incorporating higher-resolution tiled views and combining them with the upsampled feature maps from FeatUp. Additionally, FeatSharp includes a de-biasing module designed to remove fixed-pattern noise from the frozen vision encoder.

The authors find that their method out-performs FeatUp-JBU across a number of pretrained vision encoders (e.g., CLIP, DINOv2, SigLIP, SAM, RadioV2.5) for ADE20K segmentation. Additionally, they utilise their method to improve the training of a pre-existing method, AM-Radio (Multi-teacher distillation method), across a number of datasets/tasks.

[1] Fu, S., Hamilton, M., Brandt, L. E., Feldmann, A., Zhang,
Z., and Freeman, W. T. Featup: A model-agnostic
framework for features at any resolution. In The Twelfth
International Conference on Learning Representations,
2024. URL https://openreview.net/forum?
id=GkJiNn2QDF.

[2] Ranzinger, M., Heinrich, G., Kautz, J., and Molchanov, P. Am-radio: Agglomerative vision foundation model reduce all domains into one. In Proceedings of the IEEE/CVF Conference on Computer Vision and Pattern Recognition (CVPR), pp. 12490–12500, June 2024b.

**Claims And Evidence:**

The main claim that FeatSharp improves upon FeatUp-JBU is well supported. The fidelity results (Figure 5), qualitative PCA visualizations (Figure 6), and ADE20k semantic segmentation results (Figure 7) all demonstrate improvements across a variety of vision encoders. Additionally, FeatSharp is tested in a multi-teacher distillation/agglomerative model training setup ( Section 4.4) on multi-task learning benchmarks (Table 1), where it also it shown to be beneficial. The authors show the FeatSharp-trained RADIO generally performs better than the state-of-the-art RADIO-AMP-L [3].

[3] Heinrich, G., Ranzinger, M., Hongxu, Yin, Lu, Y., Kautz, J., Tao, A., Catanzaro, B., and Molchanov, P. Radio amplified: Improved baselines for agglomerative vision foundation models, 2024. URL https://arxiv.org/ abs/2412.07679

**Essential References Not Discussed:**

Not to my knowledge.

**Experimental Designs Or Analyses:**

1. Multi-view consistency / Fidelity Experiments:
This measures the mse distance between warped-upsampled features and the encoder’s low-resolution features. While this ensures consistent alignment, it doesn’t guarantee improved semantic detail. One could achieve high fidelity by over-smoothing the features so these results are generally unconvincing but do provide some additional insight to their remaining experiments.

2. The ablations given in Table 6 rely entirely on the measures of fidelity / measures of smoothness. These ablations would be more insightful if they included results using task-specific performance (eg. segmentation).

**Methods And Evaluation Criteria:**

The proposed methods and evaluation are sensible, although only segmentation results on a single dataset (ADE20k) are shown for multiple of the vision encoders (DINOv2, SigLIP, SAM). More extensive evaluation is done to show the benefit on incorporating FeatSharp within the AM-Radio method (eg. Classification: imagenet1k - Zero Shot Retrieval: COCO + Flickr30k…) but overall the methods and evaluation are reasonable.

**Other Comments Or Suggestions:**

None

**Other Strengths And Weaknesses:**

The paper is generally well-written. The technical contributions of FeatSharp are incremental, but it effectively integrates ideas from prior works and helps to build upon FeatUp and AM-Radio.

A major limitation I see is that that there are no direct comparisons made with FeatUp using implicit upsampling, which is the better performing variant of FeatUp. This is understandable to an extent as it requires training on a per-image basis, which takes ~1min/sample (Appendix G). If the performance of FeatSharp was shown to be  close/comparable to FeatUp with implicit upsampling, while greatly reducing training time this would strengthen the paper for me.

The ablations in Table 6 should show the design choice effects on some task-specific performance measures (e.g., segmentation accuracy or retrieval scores), but they instead focus on on multi-view consistency measures.

**Questions For Authors:**

My main concern is that lack of comparison against FeatUp (with implicit upsampling). The exclusion of FeatUp's implicit model is understandable given its computational cost (~1 min/sample, Appendix G). However, a small empirical comparison—at least in terms of performance and training time trade-offs—would significantly strengthen the paper's claims. Can you better justify why this is not feasible to do, at least for smaller scale experiments and add it to the paper?
Can you clarify why RADIO-AMP-L is performing worse than the baseline in Table 1?

**Relation To Broader Scientific Literature:**

The paper relates to prior work in feature upsampling, vision foundation models and multi-teacher distillation and agglomerative models. It builds directly on FeatUp [1], which introduced a model-agnostic feature upsampling framework using Joint Bilateral Upsampling (JBU) and an implicit upsampler. The authors extended the JBU variant of this work. The work is also relevant to Vision Transformers [2] and their resolution limitations due to quadratic complexity.

[1] Fu, S., Hamilton, M., Brandt, L. E., Feldmann, A., Zhang,
Z., and Freeman, W. T. Featup: A model-agnostic
framework for features at any resolution. In The Twelfth
International Conference on Learning Representations,
2024. URL https://openreview.net/forum?
id=GkJiNn2QDF.
[2] Dosovitskiy, A., Beyer, L., Kolesnikov, A., Weissenborn, D., Zhai, X., Unterthiner, T., Dehghani, M., Minderer, M., Heigold, G., Gelly, S., Uszkoreit, J., and Houlsby, N. An image is worth 16x16 words: Transformers for image recognition at scale, 2021.

**Theoretical Claims:**

Only equation (6) which appears sensible and has experimental results validating it in Appendix E.

---

> ### Author Rebuttal · Authors · 2025-03-31
>
> We thank you for your thoughtful review.
>
> > Multi-view consistency / Fidelity Experiments: This measures the mse distance between warped-upsampled features and the encoder’s low-resolution features. While this ensures consistent alignment, it doesn’t guarantee improved semantic detail. One could achieve high fidelity by over-smoothing the features so these results are generally unconvincing but do provide some additional insight to their remaining experiments.
>
> We argue that over-smoothing is actually what's happening with either bilinear upsampling, or even functionally how the JBU-stack is operating. Figure 6 shows how FeatUp 4x (JBU-stack) leads to over-smoothed results, but with edge preservation when there are strong enough edges in RGB space. This smoothness is why we compare FeatSharp against bilinear and FeatUp on fidelity directly in Figure 5, and FeatSharp consistently does better. So it's not that smoothing in the upsampled space doesn't work, but rather that it doesn't work as well as what FeatSharp is doing. Fidelity is telling us how internally consistent the representations are with respect to a particular model, under arbitrary crops and deformations. Over-smoothing would cause issues with fidelity when the crops are small, since nearly all variation will be gone.
>
> > My main concern is the lack of comparison against FeatUp (with implicit upsampling). The exclusion of FeatUp's implicit model is understandable given its computational cost (~1 min/sample, Appendix G). However, a small empirical comparison—at least in terms of performance and training time trade-offs—would significantly strengthen the paper's claims. Can you better justify why this is not feasible to do, at least for smaller scale experiments and add it to the paper?
>
> Upon your feedback, we have looked into this. It is ~1 min/sample for only the ViT-S/16 featurizer, but balloons quickly for the larger featurizers, like ~5 min/image for SigLIP2-SO400M-512, or ~4 min/image for RADIOv2.5-L/16 at 512px. However, the official implicit upsampler was also projecting the feature dimension down to 128 using PCA to achieve this speed [1, 2]. For ViT-S/16, it’s about 2x slower to use full features (e.g. 2 min/image). Due to the cost of running this mode, we ran a limited study on 100 images from COCO 2017 validation, and found that the fidelity is 3.17, higher than the 2.42 found with FeatSharp. However, FeatSharp at the same 16x upsample ratio takes 250ms/image (500x faster). These speeds are using a single H100 gpu.
>
> > Can you clarify why RADIO-AMP-L is performing worse than the baseline in Table 1?
>
> We did our best to match the training setting between our reproduction and their claimed results. A key difference seems to come down to the fact that in RADIO-AMP, they bilinear downsample the student to match the teacher in the hi-res-student/low-res-teacher setting, whereas we bilinear upsample the teacher to match the student in our work. We made this change so that we'd have a comparison with a true upsampling baseline.
>
> >The proposed methods and evaluation are sensible, although only segmentation results on a single dataset (ADE20k) are shown for multiple of the vision encoders (DINOv2, SigLIP, SAM). More extensive evaluation is done to show the benefit on incorporating FeatSharp within the AM-Radio method (eg. Classification: imagenet1k - Zero Shot Retrieval: COCO + Flickr30k…) but overall the methods and evaluation are reasonable.
>
> We have also included further benchmarking of our approach on object detection in response to reviewer oKYc below.
>
> [1] https://github.com/mhamilton723/FeatUp/blob/main/featup/train_implicit_upsampler.py#L197
>
> [2] https://github.com/mhamilton723/FeatUp/blob/main/featup/configs/implicit_upsampler.yaml#L26

---

### Decision · Program_Chairs · 2025-05-01

**Decision:**

Accept (poster)

**Comment:**

All reviewer recognise the benefit of the proposed method and recommend acceptance for the paper. Reviewers agree that the paper  presents a well-structured and comprehensive approach, demonstrating solid performance improvements across various vision tasks. I'm in favor of accepting this paper to ICML